🔓 | Open Peer Review | Bacteriology | Research Article

# Transcriptome profiling of type VI secretion system core gene *tssM* mutant of *Xanthomonas perforans* highlights regulators controlling diverse functions ranging from virulence to metabolism

Sivakumar Ramamoorthy,[1] Michelle Pena,[1] Palash Ghosh,[1] Ying-Yu Liao,[2] Mathews Paret,[2] Jeffrey B. Jones,[2] Neha Potnis[1]

**ABSTRACT**    Type VI secretion system (T6SS) is a versatile, contact-dependent contractile nano-weapon in Gram-negative bacteria that fires proteinaceous effector molecules directly into prokaryotic and eukaryotic cells aiding in manipulation of the host and killing of competitors in complex niches. In plant pathogenic xanthomonads, T6SS has been demonstrated to play these diverse roles in individual pathosystems. However, the molecular network underlying the regulation of T6SS is still elusive in *Xanthomonas* spp. To bridge this knowledge gap, we conducted an *in vitro* transcriptome screen using plant apoplast mimicking minimal medium, XVM2 medium, to decipher the effect of *tssM* deletion, a core gene belonging to T6SS-cluster i3*, on the regulation of gene expression in *Xanthomonas perforans* strain AL65. Transcriptomic data revealed that a total of 277 and 525 genes were upregulated, while 307 and 392 genes were downregulated in the mutant strain after 8 and 16 hours of growth in XVM2 medium. The transcript abundance of several genes associated with flagellum and pilus biogenesis as well as type III secretion system was downregulated in the mutant strain. Deletion of *tssM* of cluster-i3* resulted in upregulation of several T6SS genes belonging to cluster-i3*** and genes involved in biofilm and cell wall biogenesis. Similarly, transcription regulators like *rpoN*, Pho regulon, *rpoE*, and *csrA* were identified to be upregulated in the mutant strain. Our results suggest that T6SS modulates the expression of global regulators like *csrA*, *rpoN*, and *pho* regulons, triggering a signaling cascade, and co-ordinates the expression of suite of virulence factors, stress response genes, and metabolic genes.

**IMPORTANCE**   T6SS has received attention due to its significance in mediating interorganismal competition through contact-dependent release of effector molecules into prokaryotic and eukaryotic cells. Reverse-genetic studies have indicated the role of T6SS in virulence in a variety of plant pathogenic bacteria, including the one studied here, *Xanthomonas*. However, it is not clear whether such effect on virulence is merely due to a shift in the microbiome-mediated protection or if T6SS is involved in a complex virulence regulatory network. In this study, we conducted in vitro transcriptome profiling in minimal medium to decipher the signaling pathways regulated by tssM-i3* in *X. perforans* AL65. We show that TssM-i3* regulates the expression of a suite of genes associated with virulence and metabolism either directly or indirectly by altering the transcription of several regulators. These findings further expand our knowledge on the intricate molecular circuits regulated by T6SS in phytopathogenic bacteria.

**KEYWORDS**   plant-pathogen interactions, secretion system, T6SS, transcriptome sequencing, T3SS, biofilms

Address correspondence to Neha Potnis, nzp0024@auburn.edu.

The authors declare no conflict of interest.

See the funding table on p. 17.

Bacterial secretion systems are characterized into nine different groups (Tat system, types I–VII, type IX) according to their structure, function, and effector molecules that they deliver (1–4). Of these different secretion systems, the type VI secretion system (T6SS) is present in nearly 25% of Gram-negative bacteria with at least one T6SS encoded in their genome. T6SS is known to perform a wide range of functions in host-associated commensals, pathogens, as well as free-living bacteria (5–14). The T6SS is a needle-like contact-dependent molecular nanomachine that injects toxic effectors directly into cognate eukaryotic and prokaryotic cells (15). The delivered effectors subvert or manipulate host cells or fight against bacterial competitors residing in complex environmental niches (15, 16). The assembly, delivery of effectors, and disassembly of T6SS is highly dynamic. Upon induction by an unknown signal, the T6SS is assembled as a puncturing device comprising 13 essential core components, with three major proteins, TssJ, TssL and TssM, constituting the membrane complex anchored to the cytoplasmic baseplate structures. Assembly of the T6SS is a sequential process initiated by outer membrane lipoprotein TssJ, which recruits the inner membrane protein TssM anchored to another membrane protein TssL (17). TssM belongs to intracellular multiplication protein F family protein, a homolog associated with T4SSb forming a conserved T6SS component (18, 19). In *Ralstonia solanacearum*, TssM was found to play a role in virulence and secretion of proteins (20). Similarly, in *Agrobacterium tumefaciens*, TssM exhibits ATP hydrolysis function, aiding in Hcp secretion by energizing the T6SS apparatus with ATP molecules (21). In *Vibrio cholerae*, the *tssM* mutant strain formed long and pole-to-pole sheath tubes, suggesting a role in termination of sheath tube assembly rather than initiation (22). The effectors can interact specifically with VgrG proteins (23–25), the piercing device of the T6SS apparatus, or can form an effector-VgrG complex resulting in an evolved VgrG (26, 27) or conjugate with PAAR domain, which packs the toxins onto the associated VgrG (28, 29). This process is tightly synchronized by complex signaling cascade events.

Regulation of T6SS happens at different levels: transcriptional, post-transcriptional, and post-translational. Transcriptional control of T6SS genes is highly diverse among bacterial species, and numerous components, including sigma factor *rpoN* ($\sigma^{54}$) (30, 31), transcriptional regulator FleQ (32), nucleoid structuring protein H-NS (33), nutrient acquisition pathway regulators such as Fur (34), PhoB-PhoR, and secondary messenger cyclic diguanylate (c-di-GMP), have been known to regulate the activity of T6SS (32, 35, 36). Quorum sensing (QS) plays a vital role in regulating the expression of genes implicated in T6SS. For example, the T6SS was reported to be regulated by the QS system in numerous bacteria, such as *Vibrio cholerae* (37), *Vibrio fluvialis* (38), *Pseudomonas aeruginosa* (39), *Burkholderia thailandensis* (40), and *Aeromonas hydrophila* (41). Post-transcriptional regulation of T6SS genes is well characterized in *Pseudomonas* spp. The Gac/Rsm regulatory cascade controls T6SS by modulating the expression of a RNA binding protein, RsmA, which represses T6SS activity (42, 43). Two separate models have been proposed for post-translational control of T6SS gene cluster: phosphorylation-dependent and phosphorylation-independent pathways (44, 45). Since the discovery of T6SS, key breakthroughs are described in animal pathogens elucidating several mechanisms, functional attributes, and environmental cues activating T6SS (46–52). The expression of T6SS can be regulated by several environmental signals like metal ion scarcity, reactive oxygen species, pH, temperature, osmotic stress, and membrane damage (34, 53–55), but little is known on the regulation of T6SS and traits regulated by T6SS in phytopathogens.

In phytopathogenic bacteria, maintenance of a single or simultaneous multiple clusters of T6SS, their non-random distribution across phylogenetic clades, and gene flow within genera suggest the importance of this energy-expensive machinery in pathogen biology (5, 56). For example, *X. perforans* contains two T6SS clusters, i3* and i3*** (56, 57). In recent reverse genetics studies, deletion mutagenesis of core T6SS genes in different phytopathogenic bacteria revealed pleiotropic effects of T6SS on various phenotypes such as biofilm formation, motility, competition with other prokaryotic

or eukaryotic members, and virulence (10, 58, 59). Interestingly, inactivation of T6SS does not always lead to reduction in virulence even when comparing within genera. For example, mutation in *hcp* of T6SS-2 of *Xanthomonas oryzae* led to a reduction of virulence in rice, whereas our previous work indicated that mutation in *tssM* of T6SS i3* of *X. perforans* led to higher aggressiveness with faster symptom development during pathogenesis (57). Similar observations have been made, suggesting variable contributions of T6SS toward virulence in other genera including *Pseudomonas syringae* (8, 60), *A. tumefaciens* (9, 61) and *Pantoea ananatis*. Such varied contributions of T6SS in virulence in different pathosystems suggest that the role in T6SS regulation may be different, depending on the pathosystem or mode of infection. For example, *Xanthomonas euvesicatoria vgrG* and *clpV* mutants that showed no difference in virulence when compared to wild type (WT) upon infiltration directly into apoplast (62), whereas the T6SS core gene mutant of closely related sister species, *X. perforans*, showed hypervirulence on tomato upon dip inoculation (57). This work also revealed the contribution of T6SS-i3* toward osmotolerance and higher epiphytic survival in competition experiment, suggesting epiphytic fitness imparted by functional T6SS-i3* in this foliar hemibiotrophic pathogen. Similar findings highlighting the induction of T6SS during epiphytic colonization of citrus in *Xanthomonas citri* but low expression in apoplast suggest spatio-temporal regulation of T6SS during infection (63).

In the foliar hemibiotroph, *P. syringae*, upregulation of genes associated with T6SS was observed in cells exposed to increased osmotic stress, a stress commonly encountered in the phyllosphere environment (64). Another stress that pathogens experience during their colonization in the hostile phyllosphere environment is competition with the resident microbiota. In *Xanthomonas citri*, resistance to predation by the soil amoeba *Dictyostelium* was mediated by T6SS (11). In addition, the T6SS is also known to be involved in niche modification by secreting metal-scavenging proteins (65). Until now, the role played by T6SS in fitness of plant pathogens during host colonization has relied on derived phenotypic data of T6SS mutants; however, the mechanistic basis of these observations and potential direct or indirect effects on known virulence factors have not been completely deciphered. We do not know the cues involved in the activation of T6SS in phytopathogenic bacteria, whether they are limited to competitors or certain plant-derived environmental signals. Assuming that T6SS is induced in response to both cues in foliar pathogens, it is possible that gene expression patterns might have little or no overlap when comparing across individual cues. For example, T6SS might be involved in regulating genes involved in observed phenotypes associated with overall epiphytic fitness such as osmotolerance, overcoming the nutrient limitation, or competition with resident flora during early colonization events of the pathogen.

In phytopathogens where direct impact of T6SS on virulence has been validated, this may indicate involvement of T6SS in regulation of known virulence factors, such as T3SS and effectors, as the pathogen switches from epiphytic to apoplastic colonization. Thus, such spatio-temporal regulation of virulence during epiphytic or apoplastic pathogen infection makes it harder to tease apart similarities or differences in gene expression patterns in response to plant cues or microbiota cues and explain underlying phenotypes using *in planta* transcriptome screen (66). In contrast, the *in vitro* system allows discernment of these differences or similarities using controlled condition experiments. Minimal medium or nutrient-deprived synthetic medium that mimics plant apoplastic environments such as XVM2, XOM2, and XOM3 have been used previously as *an in vitro* system to trigger the virulence factors of phytopathogen (67–69). Thus, we conducted an *in vitro* transcriptome screen to understand the genome-wide expression patterns of T6SS-i3* core gene *tssM* mutant of foliar tomato/pepper pathogen, *X. perforans* (also known as *X. euvesicatoria* pv. *perforans*), under apoplast-mimicking minimal medium. Our results provide insight into the regulatory pathways modulated by T6SS which will be valuable to better understand the pathogenesis of this important phytopathogen.

## RESULTS

### Transcriptome analysis and cluster of orthologous groups-based gene expression pattern

To gain insights into the molecular mechanisms regulated by *tssM* in AL65, a genome-wide *in vitro* transcriptome analysis was performed by culturing the wild type and the mutant strains in XVM2 medium. The wild type and mutant strain were grown in *hrp*-induced medium, and total RNA was extracted at two different time points (8 and 16 h). The mRNA enriched samples were subjected to transcriptome sequencing. Analysis of RNA sequencing (RNA-Seq) data revealed that in the mutant strain, a total of 277 and 525 genes were upregulated relative to the wild type strain, while 307 and 392 genes were downregulated after 8 and 16 hours of growth in XVM2, respectively (Fig.

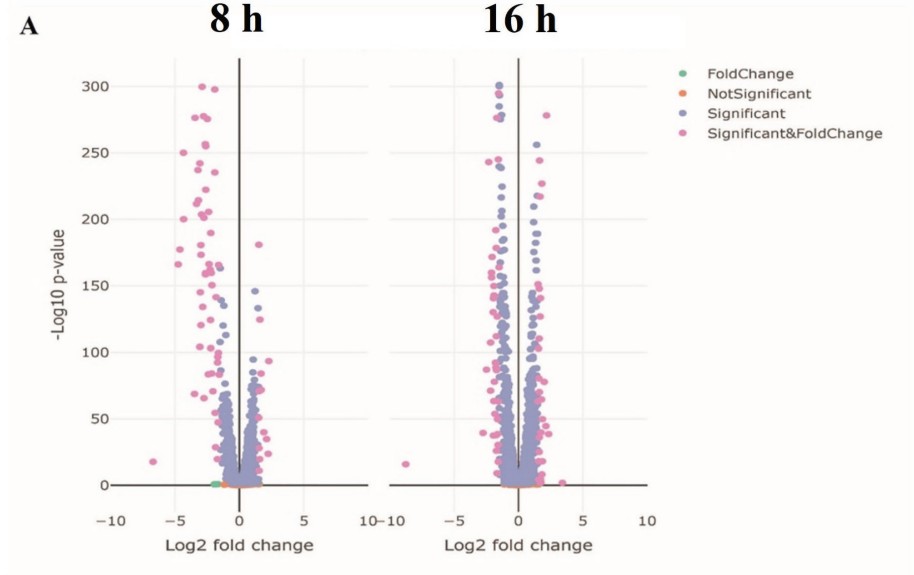

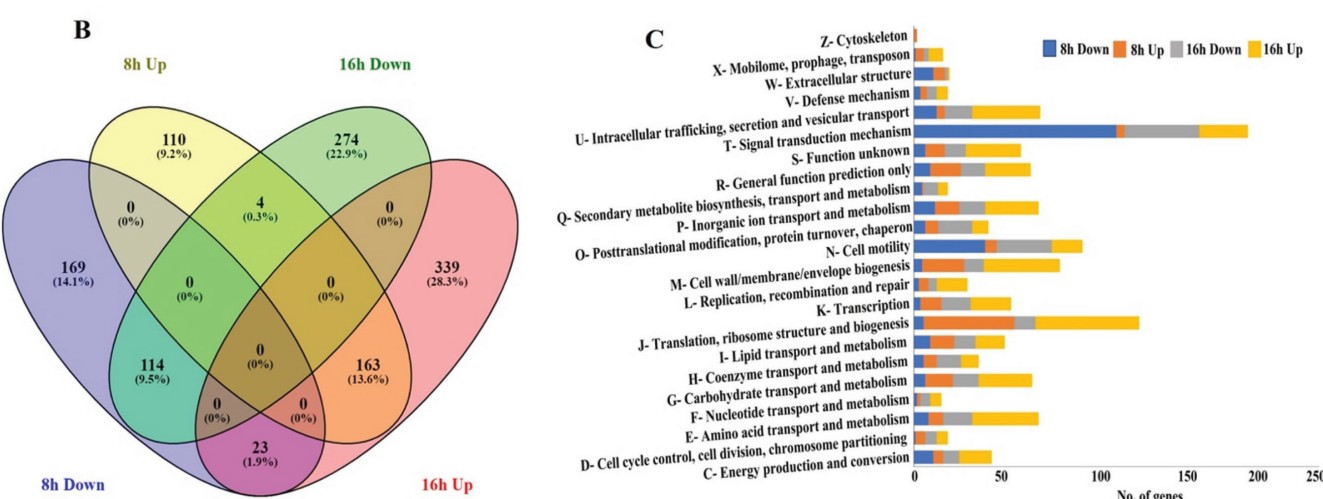

**FIG 1** *In vitro* transcriptome profiling revealed nearly 14% and 22% of genes were differentially expressed in the *tssM* mutant after 8 and 16 hours of growth in XVM2 medium. (A) Volcano plot illustrating the upregulated and downregulated genes in *tssM* mutant at 8 hours (left) and 16 hours (right). Each dot indicates an individual coding sequence with detectable expression in the mutant strain in comparison to the wild type. The *X*-axis represents the log2 fold change between mutant and wild type, and the *Y*-axis represents the –log10 transformed adjusted *P* value denoting the significance of differential expression. (B) Venn diagram of genes with significant difference in transcript level between mutant and wild type. The locus tags with significant difference in the transcript level in the mutant strain at different time points were used to construct the Venn diagram. The genes showing an overlap between the two time points are represented in percentage. (C) Functional categorization of differentially expressed genes in the *tssM* mutant based on the clusters of orthologous groups.

1A). The complete list of differentially expressed genes (DEGs) is provided in Table S1. Among the differentially expressed genes, 163 genes were upregulated at both 8 and 16 hours, and 114 genes were downregulated at both 8 and 16 hours. To check whether there was any overlap in the differentially expressed genes between the two time points, we plotted a Venn diagram. The genes showing an overlap between the two time points are represented in percentage. Interestingly, 23 genes (1.6%) downregulated at 8 hours in the mutant strain were found to be upregulated at 16 hours (Fig. 1B) (Table S2). Functional categorization of DEGs in the mutant strain showed genes associated with motility and signal transduction were highly downregulated, whereas genes implicated with ribosomal protein synthesis, cell membrane biogenesis, and intracellular trafficking and secretion were upregulated (Fig. 1C).

## Deletion of *tssM* gene resulted in downregulation of genes associated with motility and chemotaxis

We identified many genes associated with motility such as *fliDEFGHIJKLNOPS*, *flgABCDEF-GHIKLMN*, *flhFAB*, and *motABD* that were significantly downregulated in the mutant strain after *in vitro* growth in XVM2 medium for 8 and 16 hours (Table S3). These genes play a vital role in flagella-dependent swimming motility. In addition, the master regulator sigma factor FliA was found to be downregulated in the mutant strain at both time points. Further, a motility assay was performed to compare the abilities of the wild type and mutant strains to swim on a semi-solid medium. The mutant strain displayed significantly reduced swimming halos in comparison to the wild type, which was in accordance with this RNA-Seq data (Fig. 2).

The transcript level of genes involved in type IV pilus biogenesis (*pilABGMNOPQTU*) involved in major pilin and ATPase activity was reduced, while the genes associated with minor pilin (*pilEVWXY1*) were upregulated in the mutant strain at 8 hours (Table S4). The mutant strain showed reduced twitching motility in both nutrient and XVM2 medium (Fig. 3). The bulk of pilus is composed of multiple copies of a single subunit called major pilin encoded by *pilA*. Another pilin called the minor pilin found in low abundance is crucial for major pilin assembly by formation of the priming complex (70, 71). We speculate that downregulation of minor pilin in the mutant strain might affect the assembly of pilus, resulting in decreased twitching motility. Interestingly, in addition to the minor pilin, the transcript level of five genes (*pilMNOPT/U*) coding for the major pilin found to be downregulated at 8 hours displayed significant increased expression at 16 hours in the mutant strain (Table S4). Several genes involved in chemotaxis like *cheW*, *cheR*, *cheY*, *cheB*, and sensory receptor methyl accepting chemotaxis proteins displayed significant downregulation in the mutant strain at 8 and 16 hours (Table S3).

## c-di-GMP turnover genes were downregulated in the *tssM* mutant

The genome of *X. perforans* strain AL65 contains 31 genes encoding for GGDEF, EAL, and HD-GYP domains containing proteins. Among them, the transcript levels of nine genes (E2P69_RS06515, E2P69_RS12300, E2P69_RS16355, E2P69_RS12180, E2P69_RS22030, E2P69_RS10550, E2P69_RS22020, E2P69_RS01315, and E2P69_RS10910) coding for diguanylate cyclase with GGDEF domain involved in biogenesis of c-di-GMP molecule, five genes (E2P69_RS17280, E2P69_RS13670, E2P69_RS19000, E2P69_RS14210, and E2P69_RS17845) encoding for EAL domains containing protein, one gene (E2P69_RS21675) with HD-GYP domain involved in phosphodiesterase activity responsible for hydrolysis of c-di-GMP, and one gene (E2P69_RS22035) coding for the GGDEF/EAL domain containing proteins were found to be significantly downregulated in the mutant strain at 8 hours (Table S5) (Fig. 4).

## Downregulation of T3SS core genes, regulators, and effectors was observed in the mutant

The expression of T3SS genes is modulated by *hrpG* and *hrpX*, key master regulators of virulence (72). We observed that genes associated with T3SS were downregulated in the

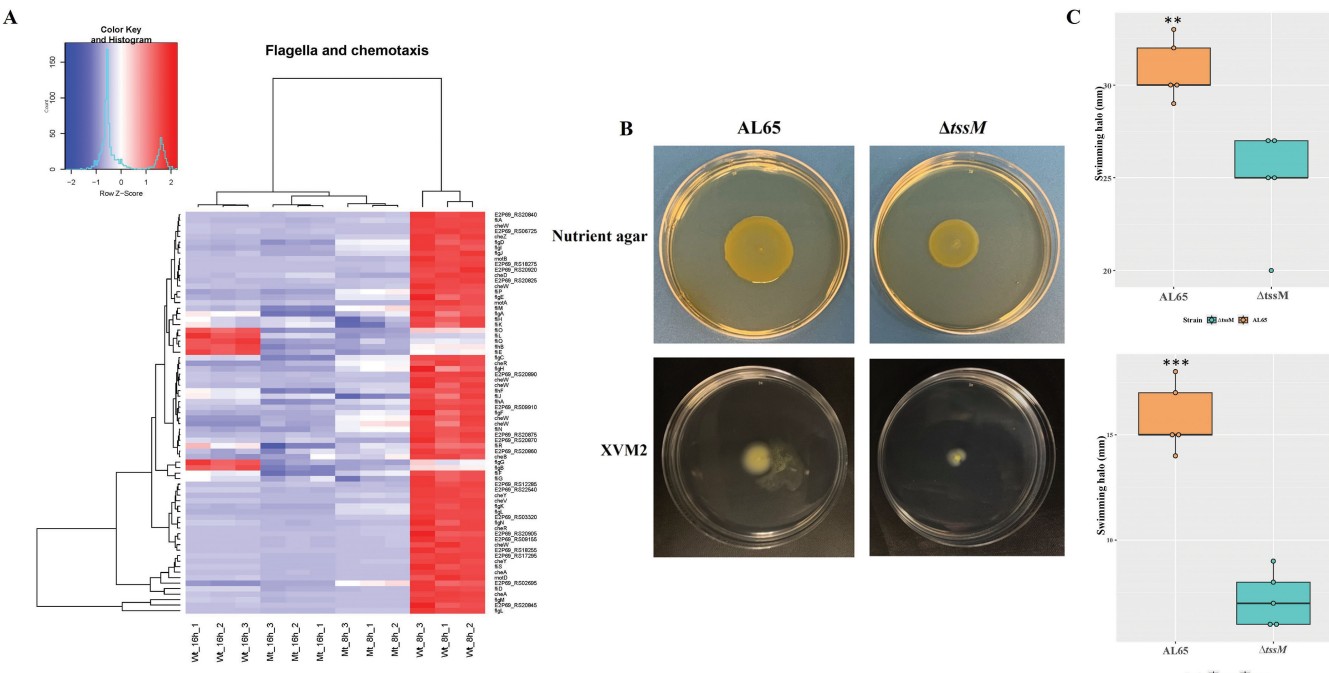

**FIG 2** Deletion of tssM resulted in downregulation of flagella biosynthesis genes. (A) Heat map showing expression pattern of genes involved in flagella production and chemotaxis. (B) Swimming motility assay for the wild type and tssM mutant. the strains were inoculated into nutrient and XVM2 containing 0.2% agar. The wild type displayed higher swimming halo in comparison to the mutant strain. The experiment was performed in five replicates, and the representative colony morphology of the wild type and mutant strains was photographed. (C) Box plot showing the diameter of the swimming halos formed by the wild type and mutant strain on nutrient agar (top) and XVM2 medium (bottom). The distribution of data were tested using Shapiro-Wilk test, and the value was greater than 0.05, showing the data were normally distributed. The significance was calculated by one-way analysis of variance. Significance: $**P < 0.01$, $***P < 0.001$. The statistics was performed using R v.23.3.0.

mutant strain at 16 hours. The expression of two genes, *hrpX* (transcriptional regulator) and *hrpG* (response regulator), was significantly downregulated by 2.6-fold (Table S6; Fig. 4). The expression of several *hrp* genes that shape the core structural apparatus of T3SS (*hrpB1*, *hrpB2*, *hrpB4*, *hrpB7*, and *hrpF*) were downregulated in the mutant strain at 16 hours (Table S6) (Fig. 4). The structure of the injectisome begins with the formation of export apparatus and the membrane rings. The RNA-Seq data revealed 10 genes, *hrcC*, *hrcJ*, *hrcL*, *hrcN*, *hpaC*, *hrcQ*, *hrcR*, *hrcS*, *hrcT*, and *hrcU*, which are implicated in the formation of export apparatus, cytosolic ring, and ATPase activity were downregulated in the mutant strain at 16 hours (Table S6). In addition to discharging virulence factors, T3SS also plays a key role in host cell sensing and modulation of gene expression post-attachment to the eukaryotic host. This function is primarily regulated by the T3SS-specific chaperone CesT/HpaB (73, 74), which was downregulated 2.8-fold in the mutant strain at 16 hours. The mRNA levels of four genes coding for type III effector proteins, HpaF, XopK, AvrXv4, and F-box, were significantly decreased between 2.5-fold and 1.8-fold in the mutant strain at 16 hours. Similarly, the transcript levels of two genes (E2P69_RS17705 and E2P69_RS22950) encoding XopP effector paralogs were downregulated by 2.4-fold in the mutant strain at 16 hours.

## *tssM* negatively regulates genes associated with T6SS under *in vitro* condition

Two T6SS gene clusters are detected in the genome of *X. perforans* strain AL65, i3* and i3***. Surprisingly, our transcriptome data revealed that deletion of *tssM*, a core gene associated with T6SS cluster i3*, resulted in upregulation of T6SS genes forming the i3* and i3*** gene clusters in the mutant strain at 16 hours (Fig. 4). Ten genes belonging to cluster i3* were upregulated in the mutant strain at 16 hours. Similarly, 10 genes belonging to T6SS cluster i3*** displayed increased expression at 16 hours (Table S7).

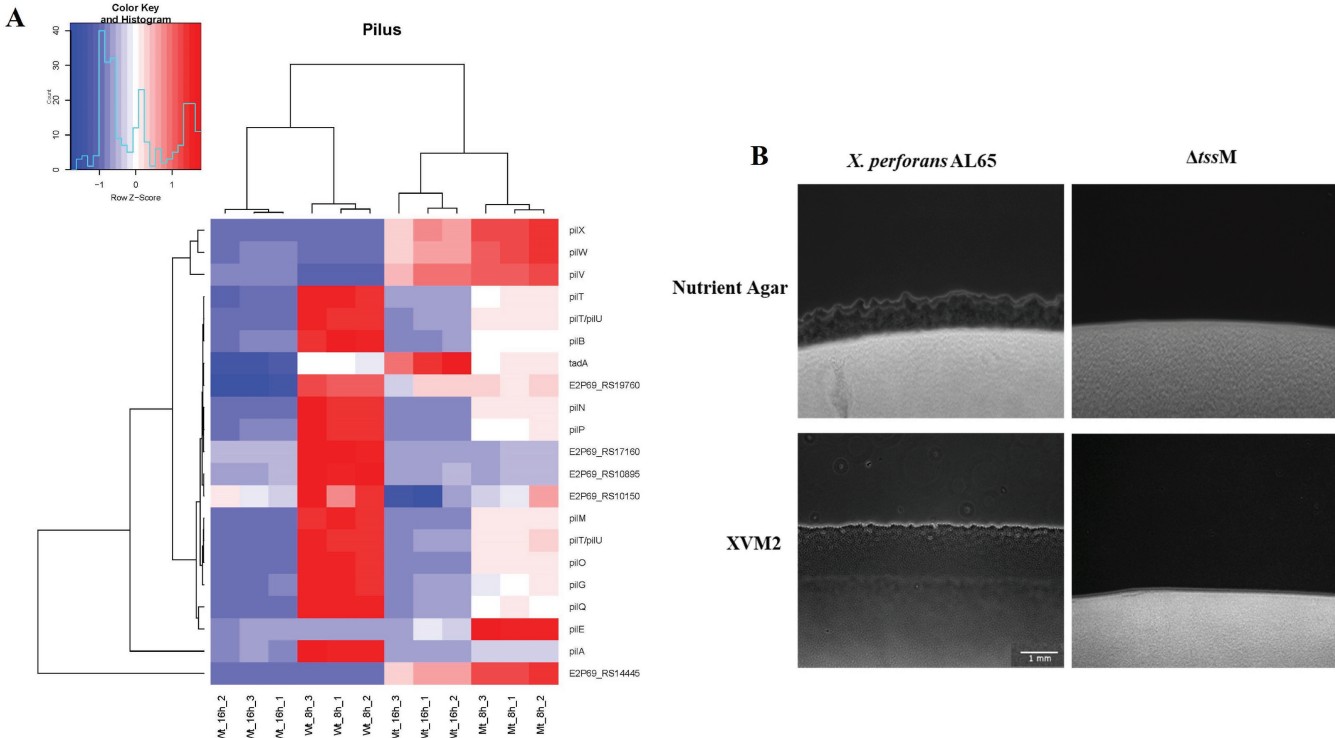

**FIG 3** Transcript level of type IV pili genes was differentially expressed in the *tssM* mutant. (A) Heat map showing expression of differentially expressed genes related to type IV pili biogenesis. (B) Colony margin characteristics of wild type and mutant strains grown on nutrient and XVM2 containing 1% agar. The wild type exhibited distinct peripheral fringe indicative of type IV pili-mediated twitching motility. Magnification scale: 1 mM.

The transcript abundance of a gene coding for OmpA family protein, which is part of T6SS accessory genes, increased in the mutant strain at 16 hours. We further identified that the expressions of seven genes (i.e., E2P69_RS05680, E2P69_RS08700, E2P69_RS17420, E2P69_RS09365, E2P69_RS19505, E2P69_RS00975, and E2P69_RS14695) that code for RHS repeat-associated core domain-containing protein, four tetratricopeptide repeat proteins, phospholipase D family protein, and cardiolipin synthase B were significantly upregulated in the mutant strain at 16 hours. These proteins constitute the T6SS-dependent putative effectors facilitating an array of interactions with prokaryotes and eukaryotes. The mRNA level of sensor kinase VirA, part of *virAG* two component system, which is known to positively modulate the expression of T6SS gene cluster in *Burkholderia* (75–77), was significantly upregulated. Post-translational (serine-threonine kinase, E2P69_RS09420; serine-threonine phosphatase, E2P69_RS09415) regulators of T6SS were identified to be upregulated in the mutant strain (Table S7). Formation of disulfide bonds in secreted effector proteins is crucial for their function and stability. In this RNA-Seq screen, the mRNA levels of two genes (E2P69_RS12845 and E2P69_RS03325) coding for thiol:disulfide interchange protein DsbA/DsbL and thiol-disulfide oxidoreductase DCC family protein displayed increased abundance in the mutant strain by 2.5-fold and 2.0-fold, respectively.

## Genes involved in T2SS, T4SS, and T5SS were differentially expressed in the *tssM* mutant

Several degradative enzymes like proteases, lipases, cellulase, and xylanases are secreted via T2SS. Genes coding for extracellular enzymes were identified to be significantly dysregulated in the mutant strain post-growth in XVM2 medium (Table S8). The mRNA level of two genes (E2P69_RS00220 and E2P69_RS00225) coding for glycoside hydrolase family 5 protein was significantly downregulated in the mutant strain at 8 and 16 hours, while one glycoside hydrolase coding gene (E2P69_RS00235) showed decreased

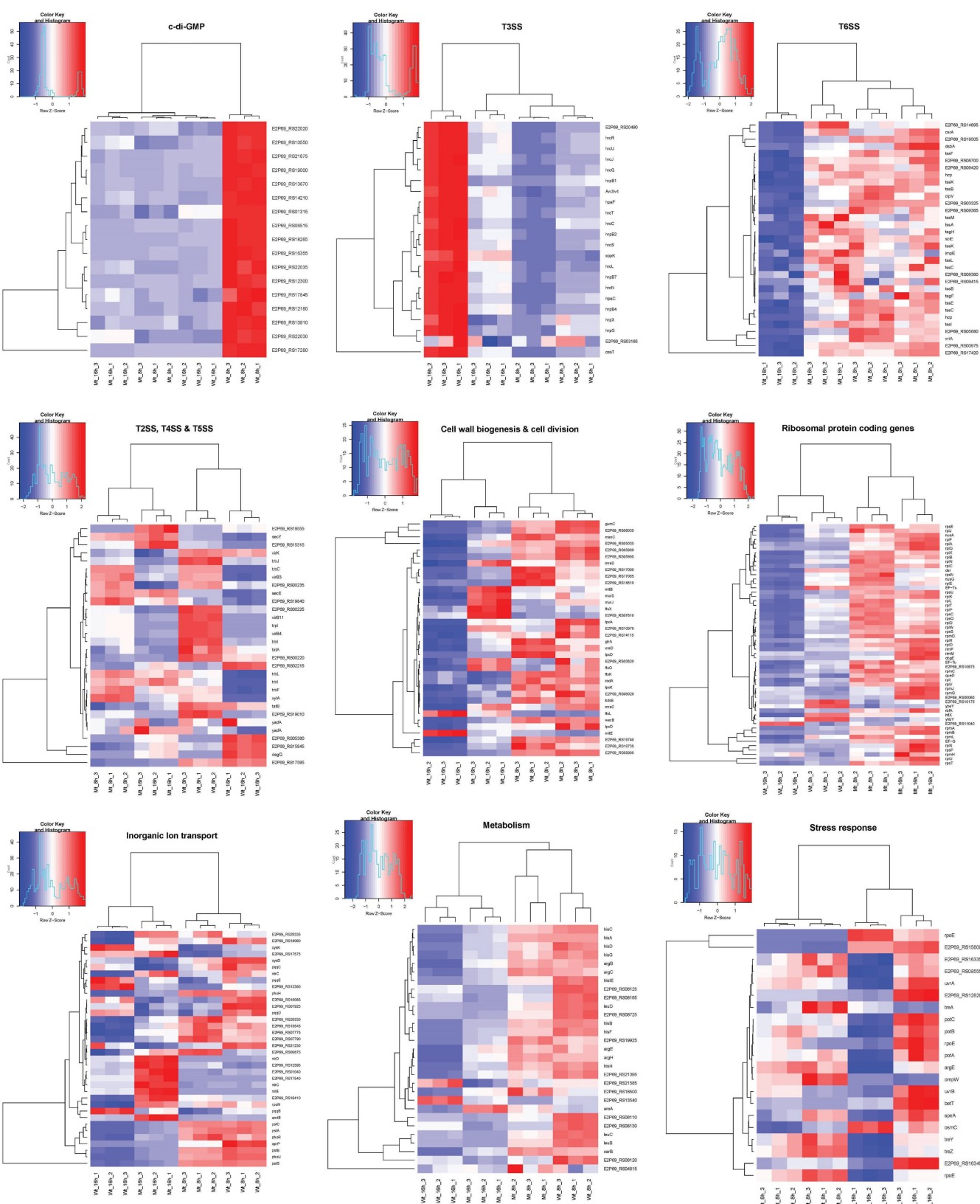

**FIG 4** Mutation of *tssM* resulted in dysregulation of genes implicated in various cellular processes. Transcriptional pattern of differentially expressed genes varying from virulence to metabolism. Heat maps showing the intensity of expression of each gene (rows) expressed as normalized read counts and scaled for each row. Each column represents the samples used in the study and is clustered based on the result of hclust analysis.

transcript level at 16 hours in the mutant strain. Blast analysis of the amino acid sequence showed this protein belongs to the BglC superfamily, which possess cellulolytic activity. Another gene (E2P69_RS19010) coding for cellulase family glycosyl hydrolase was significantly downregulated twofold in only 8 hours. The transcript abundance of two lipase coding genes (E2P69_RS02215 and E2P69_RS15945) was significantly decreased by 1.8-and 1.7- fold in the mutant strain at 16 hours. The mRNA level of xylan degradation proteins encoded by E2P69_RS05520 and E2P69_RS19555 was upregulated in the mutant strain at 16 hours, while one gene (E2P69_RS05380) coding for endo-1,4-beta-xylanase was downregulated in the mutant strain at 16 hours. The transcript level of one gene (E2P69_RS15315) coding for the alpha-amylase protein family showed an increase in expression in the mutant strain at 16 hours. In *Xanthomonas* spp., type V secretion system (T5SS) is divided into three subgroups (Va, Vb, and Vc). The expression of two genes (E2P69_RS19840 and E2P69_RS22570) coding for autotransporter domain-containing esterase and YadA-like family protein was identified to be upregulated in the mutant strain at 8 and 16 hours, respectively. These genes belong to the subgroup Va and Vc, while one gene (*fhaB*) under the Vb subgroup and two periplasmic chaperons (VirK and DegQ) were downregulated in the mutant strain. The degradative enzymes and T5SS proteins are translocated across the bacterial membrane by the Sec or TAT (twin arginine transport) pathway. Our RNA-Seq analysis revealed two Sec pathway genes (E2P69_RS22135 and E2P69_RS22465) were upregulated in the mutant strain at both time points while two genes (E2P69_RS05570 and E2P69_RS05575) belonging to TAT pathway were significantly downregulated in the mutant at 16 hours. We identified five genes (E2P69_RS14500, E2P69_RS19420, E2P69_RS14505, E2P69_RS14525, and E2P69_RS19425) associated with type IV secretion system (T4SS) in the mutant strain at 16 hours that were downregulated at 8 hours, while five genes (E2P69_RS21420, E2P69_RS21425, E2P69_RS21460, E2P69_RS21450, and E2P69_RS21445) associated with the T4SS that were upregulated in the mutant strain at 16 hours (Table S8).

## Expression of genes involved in transcription, post-transcriptional, and translational control of gene expression is modulated by *tssM*

The transcriptome data revealed that deletion of *tssM* resulted in differential expression of several genes associated with transcription, post-transcription, and translational control of gene expression. The alternative sigma factor coding for RNA polymerase factor sigma-54 (RpoN or $\sigma^{54}$) was upregulated in the mutant strain at 16 hours, which plays a role in balancing intracellular carbon and nitrogen levels. Similarly, a global regulator (*csrA*, E2P69_RS21330) associated with post-transcriptional regulation of gene expression was upregulated in the mutant strain at both 8 and 16 hours (Table S9). RNase plays a vital role in post-transcriptional modulation of gene expression in pathogenic bacteria aiding the organism to swiftly adapt to the host environment. In the transcriptome analysis, we identified the mRNA level of two genes (E2P69_RS00065 and E2P69_RS10975) coding for ribonuclease *P* protein component, and ribonuclease HII was significantly upregulated in the mutant strain at 8 and 16 hours, while two genes (E2P69_RS10175 and E2P69_RS09245) coding for ribonuclease PH and rRNA maturation RNase YbeY were significantly downregulated in the mutant strain at 8 hours.

Several genes coding for ribosomal subunit protein were identified to be highly upregulated in the mutant strain at both time points (Table S9) (Fig. 4). The ribosomal proteins constituted 26% and 12% of upregulated genes in the mutant strain at 8 and 16 hours post-growth in the XVM2 medium. The composition of bacterial ribosomal proteins is highly heterogenous and has paralogs of some ribosomal proteins that have evolved to combat environmental stress. For example, the transcript abundance of two genes (E2P69_RS10215 and E2P69_RS08540) coding for type B 50S ribosomal protein L31 and L36 was significantly higher in the mutant strain at 8 and 16 hours. These genes play a vital role in alleviating zinc starvation stress (78). In addition to genes involved in ribosomal protein biogenesis, the transcript levels of several genes implicated in maturation and assembly of ribosomes were significantly increased in the mutant

strain. Ribosome assembly factors like GTPase (ObgE, Der, and HflX), maturation proteins (RimM and RimP), and protein chaperone (RbfA) were upregulated at both time points, whereas the transcript abundance of one gene (E2P69_RS07135) coding for RNA-binding protein YhbY associated with ribosome assembly was decreased in the mutant strain at 8 hours. The precision and efficiency of protein synthesis can be modulated by tRNA modification. Queuosine tRNA modification is widespread in prokaryotes, which play a role in efficacy and accuracy of translation. The mRNA level of a gene (E2P69_RS04970) involved in queuosine metabolism was significantly upregulated 2.2-fold in the mutant strain at 8 hours. One gene coding for ribosome-associated translation inhibitor RaiA was identified to be significantly downregulated in the mutant strain at 16 hours. This gene plays an important role in ribosome hibernation, resulting in loss of translational activity of the ribosomes. Indeed, this downregulation of translation inhibitor aligns with the observation of significant upregulation of various ribosomal subunits encoding genes in the mutant.

## Deletion of *tssM* resulted in upregulation of genes involved in cell wall biogenesis

Several genes coding for cell wall biogenesis were found to be significantly upregulated in the mutant strain. Nine genes coding for glycosyltransferases (E2P69_RS06005, E2P69_RS19735, E2P69_RS06015, E2P69_RS05035, E2P69_RS19740, E2P69_RS05995, E2P69_RS05990, E2P69_RS17085, and E2P69_RS07810) were upregulated at 16 hours (Table S10). These enzymes are essential for the formation of repeating units of monosaccharide moieties, which are the basic units for biosynthesis of peptidoglycan, exopolysaccharides (EPSs), and lipopolysaccharides (LPSs). Genes involved in the biosynthesis of LPS displayed significantly increased expression in the mutant strain. Two genes (E2P69_RS10970 and E2P69_RS03820) coding for O-antigen ligase family protein and lipid-A-disaccharide synthase were found to be upregulated at 8 and 16 hours, while one gene (E2P69_RS05050) involved in O-antigen modification was upregulated only at 16 hours. The transcript level of three genes (E2P69_RS22825, E2P69_RS08060, and E2P69_RS08055) coding for tetraacyldisaccharide 4′-kinase LpxK and mannose-1-phosphate guanylyltransferase/mannose-6-phosphate isomerase, 3-deoxy-manno-octulosonate cytidylyltransferase was upregulated at 16 hours. These genes are crucial players in the biosynthesis of LPS. Four genes associated with EPS production coding for glycosyltransferase (E2P69_RS06005), acyltransferase family (E2P69_RS05980), *gum* cluster cupin domain (E2P69_RS06020), and *gumC* family protein (E2P69_RS05965) part of *gum* cluster were identified to be significantly upregulated in the mutant strain at 8 hours. *In vitro* assays were conducted to quantify the biofilm and exopolysaccharide production ability of wild type and mutant strains. There was no significant difference in the biofilm forming ability between the wild type and the mutant strains. The *in vitro* assay revealed increased EPS present in the *tssM* mutant at 48 hours, although the difference was not significant (Fig. S1). Four genes encoding for UDP-3-O-(3-hydroxymyristoyl) glucosamine N-acyltransferase, acyl-ACP-UDP-N-acetylglucosamine O-acyltransferase, polysaccharide pyruvyl transferase family protein, and murein biosynthesis integral membrane protein MurJ were found to be upregulated in the mutant strain at 16 hours (Table S10) (Fig. 4). These proteins play a major role in biosynthesis of peptidoglycan. In addition, genes implicated in remodeling of cell walls for anchoring surface appendages like flagella, pili, and channels for translocation of DNA, nutrients, and protein effectors across membrane were also found to be differentially expressed in the mutant strain. A transglycosylase (E2P69_RS14115) encoding gene was upregulated in the mutant strain at 8 and 16 hours, while a gene coding for lytic murein transglycosylase MltB (E2P69_RS14605) was upregulated only at 16 hours. The expression of two genes (lytic transglycosylase domain-containing protein MltE and flagellar assembly peptidoglycan hydrolase FlgJ) was downregulated in the mutant strain at 8 and 16 hours. The latter gene *flgJ* is essential for the assembly of flagellar apparatus on the bacterial cell membrane.

# The expression of metabolic and inorganic ion transport genes is differentially regulated by *tssM* in AL65 under *in vitro* condition

Several genes involved in nitrogen metabolism were significantly upregulated in the mutant strain. The expression of other nitrogen-regulated genes under *rpoN* regulon like *narA*, *nirBD*, *ntrC*, and *glnK-amtB* was significantly increased in the mutant strain (Table S11). Like nitrogen metabolism, several genes implicated in phosphate (Pi) homeostasis were found to be significantly upregulated in the mutant strain. One gene coding for sensor hisdine kinase PhoR, part of two-component system (TCS) PhoBR, was upregulated in the mutant strain at 16 hours. The transcription of genes under Pho regulon is modulated by PhoBR in response to external Pi concentration. Similarly, the transcript abundance of a gene coding of PhoH family protein belonging to the core member of Pho regulon was significantly increased by 1.56-fold in the mutant strain at 16 hours. In addition to the TCS PhoBR, Pi homeostasis is also maintained by the Pst system. In our RNA-Seq analysis, all four genes of Pst operon (*pstSCAB*) were found to be significantly upregulated in the mutant strain at 16 hours. The phosphate scavengers like phytase (E2P69_RS20335), alkaline phosphatase (E2P69_RS18060) and glycerophosphodiester phosphodiesterase (E2P69_RS16410) were upregulated in the mutant strain at 16 hours. Four genes (E2P69_RS02800, E2P69_RS02810, E2P69_RS02805, and E2P69_RS02815) coding for pyrroloquinoline quinone biosynthesis protein PqqB, pyrroloquinoline quinone biosynthesis peptide chaperone PqqD, pyrroloquinoline-quinone synthase PqqC, and pyrroloquinoline quinone biosynthesis protein PqqE involved in phosphate solubilization were downregulated in the mutant strain at 16 hours. One gene coding for a phosphate specific porin (E2P69_RS15410) was downregulated in the mutant strain at 8 hours. Several genes involved in sulfur transport and metabolisms were identified to be differentially expressed in the mutant strain. Three genes (E2P69_RS07775, E2P69_RS12595, and E2P69_RS07790) associated with sulfur uptake were upregulated at 8 and 16 hours, while four genes (E2P69_RS13380, E2P69_RS07925, E2P69_RS20745, and E2P69_RS20680) coding for TauD/TfdA family dioxygenase, sulfurtransferase, cysteine synthase A, and sulfate adenylyltransferase subunit CysD were downregulated in the mutant strain at 16 hours.

Transcriptome analysis revealed differential expression of several metabolic genes in the mutant strain. The transcript level of 10 genes (*hisA-B-C-D-F-G-H-IE*) involved in histidine biogenesis was significantly upregulated in the mutant strain at 16 hours (Table S12). *De novo* synthesis of histidine plays an essential role in cellular metabolism. Similarly, five genes (E2P69_RS08790, E2P69_RS08775, E2P69_RS08795, E2P69_RS08780, and E2P69_RS21025) associated with the arginine biosynthesis pathway were significantly upregulated in the mutant strain at 16 hours. Arginine biosynthesis was considered a virulence trait as demonstrated by pathogenesis studies in Xanthomonadaceae species (Sagawa et al., 2022) (79) . Three genes (E2P69_RS07515, E2P69_RS19925, and E2P69_RS16500) coding for 3-phosphoshikimate 1-carboxyvinyltransferase, 3-deoxy-7-phosphoheptulonate synthase, and aminodeoxychorismate synthase component I were upregulated in the mutant strain at 8 and 16 hours, while one gene (E2P69_RS04815), which encodes for chorismate mutase, was upregulated at 8 hours in the mutant strain. These genes are important for aromatic amino acid biosynthesis. The transcript abundance of genes associated with the leucine biosynthesis pathway was significantly downregulated in the mutant strain. Two genes (E2P69_RS09875 and E2P69_RS09870) coding for 3-isopropylmalate dehydrogenase and 3-isopropylmalate dehydratase small subunit were downregulated at 16 hours, while one (E2P69_RS09865) encoding for 3-isopropylmalate dehydratase large subunit was downregulated in the mutant strain at 8 and 16 hours (Table S12). In addition to amino acid metabolism, the mRNA levels of genes involved in carbohydrate transport and metabolism were identified to be differentially expressed. Six genes (E2P69_RS08130, E2P69_RS08125, E2P69_RS08120, E2P69_RS08110, E2P69_RS08725, and E2P69_RS08195) encoding for succinate dehydrogenase cytochrome b556 subunit, succinate dehydrogenase hydrophobic membrane anchor protein, succinate dehydrogenase flavoprotein subunit,

succinate dehydrogenase iron-sulfur subunit, cytochrome ubiquinol oxidase subunit I, and c-type cytochrome were downregulated in the mutant strain at 8 and 16 hours, whereas the transcript level of genes coding for cytochrome c maturation and cytochrome b biogenesis was upregulated in the mutant strain. These genes play an important role in electron transport chain and energy production (Table S12). The transcript level of two genes (E2P69_RS13540 and E2P69_RS21585) encoding for protocatechuate 3,4-dioxygenase subunit alpha and protocatechuate 3,4-dioxygenase subunit beta was lower at 16 hours in the mutant strain. These proteins play an essential role in the degradation of 4-hydroxybenzoate, an intermediate used in xanthmonadin pigment synthesis (80).

## Deletion of *tssM* results in upregulation of genes involved in stress response

Two genes (E2P69_RS16345 and E2P69_RS16335) involved in α-glucan biogenesis were upregulated in the mutant strain at 16 hours (Table S13). Similarly, two genes (E2P69_RS16330 and E2P69_RS16340) responsible for trehalose biosynthesis were upregulated in the mutant strain at 16 hours, while one gene coding for alpha, alpha trehalase TreA, involved in degradation of trehalose was found to be downregulated in the mutant strain at 8 hours. These genes are essential for alleviating abiotic stress. Polyamines play an important role in DNA replication, RNA translation, and cell membrane integrity by binding to polyanionic compounds. The major polyamines found in bacteria are putrescine and spermidine. Our RNA-Seq analysis revealed six genes (E2P69_RS08795, E2P69_RS12820, E2P69_RS15510, E2P69_RS12825, and E2P69_RS08555) involved in the synthesis of putrescine and spermidine were upregulated in the mutant strain at 16 hours. Three genes (E2P69_RS09280, E2P69_RS09275, and E2P69_RS06565) coding for ABC transporter permease PotA, PotB, and PotC involved in exchange of polyamines were found to be upregulated in the mutant strain. Similarly, two genes (E2P69_RS14465 and E2P69_RS11830) coding for excinuclease ABC subunit UvrB and excinuclease ABC subunit UvrA were upregulated in the mutant strain. The mRNA level of a general stress protein CsbD (E2P69_RS15800) was significantly increased by 2.2-fold in the mutant strain at 8 hours. The transcript abundance of two genes (E2P69_RS10375 and E2P69_RS01815) coding for OsmC family protein and OmpW family protein was downregulated in the mutant strain at 16 hours by 1.8-fold and 1.5- fold, respectively. Interestingly, we identified two genes (E2P69_RS15710 and E2P69_RS09345) coding for sigma factors RpoE that were upregulated in the mutant strain at both time points. This sigma factor plays a crucial role in stress response in bacteria (Table S13).

## DISCUSSION

T6SS is a multifunctional protein firing apparatus in Gram-negative bacteria. Although genes associated with T6SS have been identified in many species of *Xanthomonas*, an important plant pathogen, the biological and molecular functions are not completely understood in many xanthomonads (14). Previously, we reported deletion of *tssM*, a core T6SS i3* gene, resulted in hypervirulent phenotype of *X. perforans* strain AL65 (57). To characterize the regulatory circuit modulated by TssM, we performed a transcriptome analysis by growing the wild type and the mutant strain in an *hrp*-inducing medium, XVM2. Based on our findings, TssM of T6SS-i3* directly or indirectly controls the transcription level of several genes that participate in bacterial virulence (motility, secretion system, and biofilm), stress response, and metabolic pathway (amino acid and carbohydrate metabolism) through a regulatory network (Fig. 5), indicating its pleiotropic effects.

In the *tssM* mutant, transcript level of flagellar biogenesis genes was downregulated, revealing that functional TssM positively regulated these determinants under the tested condition. Flagella is essential for motility and chemotaxis of bacterial cells. The contribution of flagella in virulence and pathogen-associated molecular pattern was established in numerous pathogenic bacteria, including *X. campestris* pv. *campestris* (88–91).

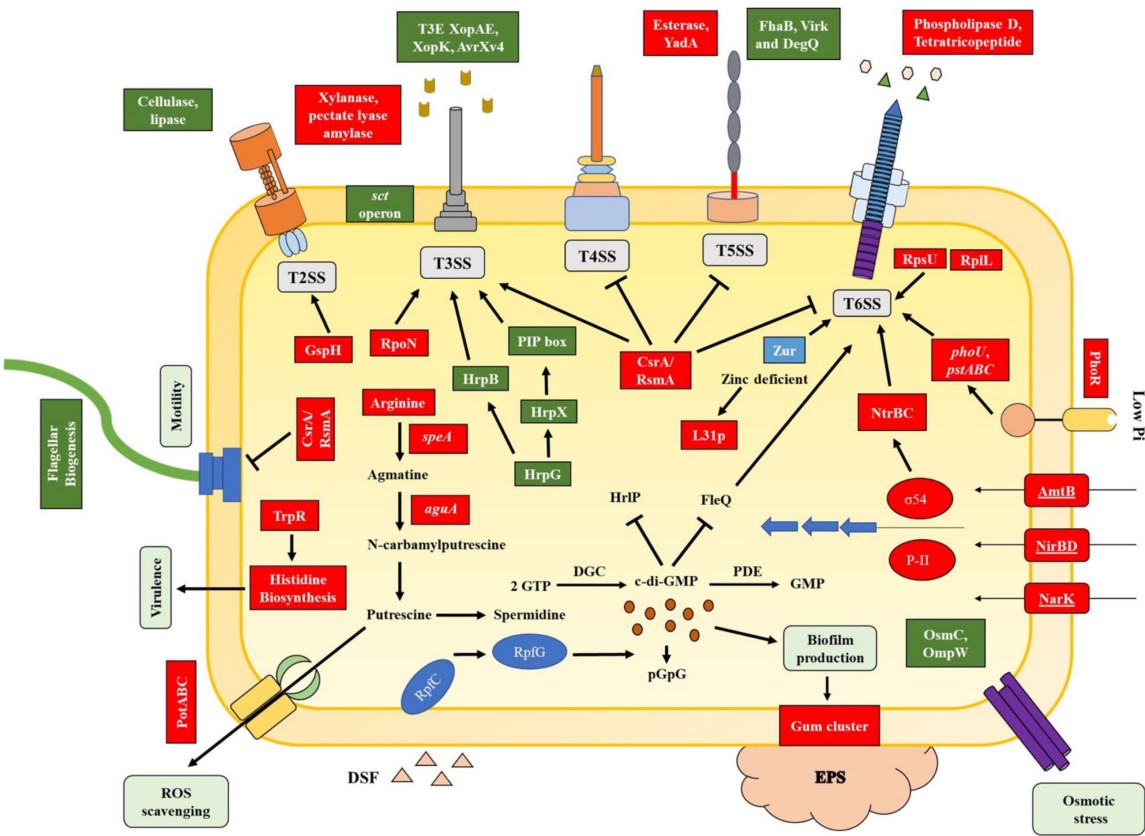

**FIG 5** Model illustrating the regulatory cascade modulated by tssM in AL65 under *in vitro* condition. The proposed model provides an overview into the molecular regulation of different functions in tssM mutant ranging from virulence to metabolism controlled by major regulators identified in the present study. One major regulator of virulence factors is the nucleotide secondary messenger c-di-GMP. The intracellular levels of c-di-GMP negatively regulate flagella and T3SS (81, 82) while having a positive effect on biofilm formation. The higher transcript level of gum cluster genes positively regulates the production of exopolysaccharide. In *Xanthomonas* sp., the secretion and delivery of toxic effectors via T3SS are controlled by two major transcriptional regulators HrpG and HrpX (83). Phosphorylated HrpG binds directly to the promoter region of hrpX. HrpX induces the expression of T3SS and T3Es coding genes by binding to the plant inducible promoter (PIP) box, a DNA motif found in the structural T3SS operon and TSE gene. The RNA binding protein, CsrA, positively regulates T3SS while repressing the expression of T4SS, T5SS, T6SS, and flagella biogenesis genes. FleQ positively controls flagella biosynthesis and negatively regulates T6SS genes (32). The available nitrate from the extracellular space is imported into the cytoplasm by nitrogen-related transporter and reductase coding genes (narA, nirBD, nark, and amtB), which are regulated by the master regulator RpoN (σ54). RpoN positively regulate the expression of T3SS and T6SS genes. Under phosphate-limited condition, the histidine kinase PhoR interacts with the PstABC-PhoU complex and results in phosphorylation of PhoB, which positively regulate the expression of T6SS genes (84, 85). At low concentration of zinc, the Zur regulator induces the expression of ZnuABC and activates T6SS genes (86). Breakdown of arginine by catabolic enzymes (SpeA and AguA) produces putrescine, exported out of the cell by transporters (PotABC) and helps in ROS scavenging (87).

Similarly, TssM positively regulates the expression of c-di-GMP turnover genes. In this study, several genes involved in synthesis and degradation of c-di-GMP were identified to be downregulated in the mutant. Since genes coding for both GGDEF and EAL/HD-GYP domain containing proteins were downregulated in the mutant strain, further quantification of the intracellular levels of c-di-GMP in the mutant and the wild type strains would help to understand the dynamic mechanistic framework underlying regulation of virulence factors by T6SS via c-di-GMP.

In the present study, the expression of genes involved in T3SS and T3E was downregulated in the mutant strain, which was intriguing, given the observation of hypervirulent phenotypes with mutant *in planta* (57). We speculate one reason for decreased transcription level of T3SS genes could be the existence of a putative crosstalk between T6SS and T3SS as reported previously. Deletion of single T6SS resulted in decreased expression of T3SS genes in *Erwinia amylovora* (92) and *Pseudomonas syringae* pv.

*actinidiae* (93). Recently, Li et al. (94) reported similar results in XVM2 medium, where the expression of *hrp* genes was decreased in *Xanthomonas campestris* pv. *campestris* (*Xcc*) *hpaP* deletion mutant. HpaP (HR and pathogenicity associated phosphatase) is a regulatory protein with both ATPase and phosphatase activity required for complete virulence in *Xcc* regulating the expression of *hrp* genes (95). However, a contrasting expression pattern was observed when the pathogen was infected to the plant. Although, the minimal medium, XVM2, which is suspected to mimic the plant apoplastic environment, was used as an alternative system for transcriptome expression studies under controlled conditions, it lacks specific plant signals that trigger the *hrp* regulon *in vitro*. The induction of T3SS genes is much stronger in the presence of plant-derived signals than in minimal medium (96, 97). In *P. syringae* DC3000 strain, exposure to tomato exudates activates the expression of *hrpK*, suggesting that the phytopathogen senses the soluble low-molecular weight host signals in the exudate resulting in activation of T3SS genes (98). However, the mechanism by which the pathogen senses the plant signal and the regulatory mechanism that contributes to the induction of host factors and virulence factors in pathogens is unclear. This result and our findings imply that the mechanism of *hrp* genes activation and regulation under *in vitro hrp*-inducing conditions is divergent from that *in planta*. Regardless, the present study provided us with the understanding of the regulatory network of TssM without the complexity of spatial regulation observed *in planta*. Under such simplified conditions, inactivation of TssM leading to decreased T3SS gene expression suggest intertwined network with direct or indirect interaction of T6SS and T3SS.

One striking result observed in this RNA-Seq analysis was the significant upregulation of T6SS genes belonging to both clusters i3* and i3*** in the *tssM* mutant. In *Vibrio cholerae*, constitutive expression of T6SS results in feedback control by sensing the Hcp quantity in the cytosol. The increased amount of Hcp in the cell leads to downregulation of T6SS genes. In a *tssM* mutant background, the mRNA level of *hcp* was found to be downregulated, but the expression level of other core genes was unaltered (99). In our study, we observed a contradicting result where the expression of *hcp* was upregulated in the *tssM* mutant. Another indicative of functional T6SS is secretion of *hcp* out of the cell. We speculate that although the *hcp* gene expression of i3* cluster was upregulated in the mutant, it is not secreted into the supernatant. Also, the physical separation of the two clusters into distinct operons may be suggestive of different transcriptional regulations. The presence of multiple T6SS has been reported in several plant pathogens, and each could be regulated differently and might carry out diverse functions as evident in other phytopathogens. In *X. citri*, T6SS-i3* was induced in response to predation by soil-dwelling amoeba *Dictyostelium discoideum* (11). The expression of T6SS-i3* was regulated by a series of signaling cascade events modulated by a serine/threonine kinase, which was also upregulated in our study. This is important as the phyllosphere serves as a habitat for bacterial predators; induction of T6SS-i3* helps to compete for niche during epiphytic colonization. In contrast, T6SS-i3*** subgroup of *X. oryzae* pv. *oryzicola* was induced within the plant host revealed by a transcriptome study, but its function remains unclear (100). Of the two T6SSs identified in *Pantoea ananatis* LMG 2665, a major pathogen of pineapple and onion, T6SS-1 was found to be responsible for pathogenicity and competitiveness, while T6SS-3 had no effect on virulence of the strain (10). Similarly, mutants in T6SS-1 and T6SS-2 genes of *Xanthomonas oryzae* pv. *oryzicola* resulted in no significant difference in virulence when compared to the wild type but was required for bacterial competition (14). It is still elusive how various pathways integrate to activate the T6SS machinery (101). Overall, this supports the notion that different phytopathogens have specific regulatory networks to deploy their T6SS in response to different environmental signals. Given our observation of increased expression of genes from both T6SS clusters may indicate a crosstalk among the two clusters.

Many genes associated with polysaccharide production and its regulation were highly upregulated in the mutant strain. High-molecular weight extracellular polysaccharides and glycoproteins are key components of the bacterial cell wall. In pathogenic bacteria,

these molecules are crucial players in evading host immune responses, colonization, stress response, and biofilm formation (102, 103). In *Xylella fastidiosa*, O-antigen delays host immune response in grape xylem (104). Similarly, several genes with glycosyltransferase were essential for fitness of *P. syringae* during leaf and apoplastic colonization (105). We reasoned that the increased expression of polysaccharide could be attributed to the increased expression of the global regulator *csrA* in the mutant strain. In xanthomonads, production of exopolysaccharide xanthan gum is positively regulated by CsrA (106), which contributes to proliferation of bacteria within intracellular apoplastic space, resulting in development of disease symptoms (107, 108). The regulatory network of CsrA/RsmA is poorly understood in *Xanthomonas* spp. It is interesting that RsmA/CsrA negatively regulates genes associated with flagellum biogenesis, T4SS, T5SS, and T6SS, which are essential for epiphytic colonization of pathogen while positively regulating T3SS genes required for apoplastic colonization. This may imply that CsrA could possibly function as a molecular switch in *Xanthomonas* spp., shifting from a epiphytic to an endophytic lifestyle. Another major regulator, the alternative sigma factor RpoN that influences biofilm formation, virulence, and growth displayed increased expression in the mutant strain (109–113). The role of RpoN in regulating T6SS is well established in animal pathogens (30, 114, 115), but in phytobacteria, very little is known, which opens an area for further investigation.

During colonization of the host, the pathogen is constantly exposed to abiotic stresses. The ability to resist these stresses plays a vital role in the fitness of the organism. In this transcriptome analysis, transcript abundance of several genes implicated with polyamine biosynthesis and transport was significantly upregulated in the mutant strain. In phytopathogens, synthesis and secretion of putrescine was reported in *R. solanacearum*, *P. syringae*, and *Dickeya zeae* (116–118), which play an important role in alleviating the oxidative stress by scavenging the reactive oxygen species (119, 120). Further, two genes (*osmC* and *ompW*) associated with osmotic stress were significantly downregulated in the mutant strain, which was in concordance with our previous result in which the *tssM* mutant displayed reduced growth when exposed to osmotic stress (57). These findings suggest the role of functional TssM in stress tolerance.

In summary, our data demonstrate that a highly intricate signaling pathway orchestrated by multiple regulators is involved in the regulation of T6SS genes. We observed that expression of major virulence factor regulators like c-di-GMP metabolic genes, sigma factors (RpoN, RpoE, and FliA), inorganic ion homeostasis genes (Pho regulon and Zur), and RNA-binding protein CsrA was controlled by TssM either directly or indirectly, as shown in the schematic representation of the regulatory circuit mediated by *tssM* gene (Fig. 5). Based on our study and from previous works, we conclude that regulatory networks modulating virulence factors are highly complex in *Xanthomonas*, and crosstalk among T6SS and other secretion systems may govern virulence, stress tolerance, and metabolism. Our findings also highlight the importance of *in vitro* transcriptomics to understand basal regulatory networks that are otherwise highly context dependent, be it host-derived cues or spatial expression.

## MATERIALS AND METHODS

### Bacterial strains and growth conditions

*Xanthomonas perforans* WT strain AL65 and its isogenic *tssM* mutant were routinely cultured on nutrient agar (NA) (Difco, USA) plates at 28°C. Liquid cultures were grown to the exponential phase at 28°C with continuous shaking at 150 rpm. For the RNA-Seq experiment, *X. perforans* strains were grown in XVM2 medium [10-mM sucrose, 10-mM fructose, 0.01-mM casamino acids, 10-mM $(NH_4)_2SO_4$, 5-mM $MgSO_4$, 0.16-mM $K_2HPO_4$, 0.32-mM $KH_2PO_4$, 20-mM NaCl, 1-mM $CaCl_2$, 0.01-mM $FeSO_4$, pH 6.7] (121) with initial inoculum concentration of $10^5$ CFU/mL. Both wild type and the mutant strains were

grown at 28°C with agitation. The bacterial cells were collected post-growth in XVM2 medium at 8 and 16 hours post-inoculation.

## RNA extraction

Total RNA was extracted using TRIzolTM (Thermo Fisher Scientific, Waltham, MA, USA) protocol (122). Four independent RNA extractions were obtained for each strain for each time point. The concentration of RNA samples was determined using QubitRNA HS Assay kit (Invitrogen). Ribosomal RNA was depleted using the MICROB Express Bacterial mRNA purification kit (Ambion, Foster City, CA, USA). The integrity and quality of the RNA were checked by an Agilent 2100 Bioanalyzer (Agilent Technologies, Santa Clara, CA, USA).

## RNA-Seq data processing and differential expression analysis

The RNA sequencing was performed at MiGS using Illumina NextSeq 2000 to generate 12-M, $2 \times 51$ bp reads/sample. The sequenced data were processed to generate FASTQ files. Two pipelines were used to calculate differential gene expression (DGE) to account for variation observed depending on models used for transcript estimation (123). The first pipeline used was epinatel (https://github.com/epinatel/Bacterial_RNAseq/). The second pipeline used for DGE analysis was brnaseq (https://github.com/barricklab/brna-seq/blob/master/README.md). Both pipelines used steps involving read quality filtering using trimmomatic (124), mapping to the reference genome of *X. perforans* AL65 (RefSeq: NZ_SMVI00000000.1, assembly: GCF_007714115.1) using bowtie2 (125), DGE using DESeq2 (126), except for read counting step with the first pipeline using custom scripts and the second using HTSeq (127). Significantly upregulated or downregulated genes were identified as those in which the absolute value of log (in base 2) of fold change was greater than 0.58, and the adjusted false discovery rate was ≤0.05. The output of two pipelines was compared. The data obtained from epinatel are represented in tables and figures (after confirming no difference in top DEGs obtained in both methods). Gene ontology analysis was performed using EggNOG v.5.0 (128).

## Motility

The ability of wild type and mutant strains to swim and twitch was screened on NA and XVM2 agar. Single colonies of *X. perforans* AL65 wild type and AL65Δ*tss*M mutant were inoculated into nutrient broth and XVM2 medium and grown for 24 hours at 28°C. Bacterial growth was then normalized using respective media to $10^8$ CFU/mL ($OD_{600 nm}$ = 0.3) using spectrophotometer (GENESYS 10S UV-Vis, Thermo Fisher Scientific). For each strain, 2 µL of the suspension was placed onto the center of the plates and incubated at room temperature for 48 hours without agitation. The swimming halos were measured to identify the difference in the motility between wild type and the mutant strain. For twitching motility, the fringe width was visualised using a microscope (Revolve Microscope; Echo Laboratories, San Diego, CA, USA). The experiment was performed in triplicate.

## Biofilm assay

The wild type and the mutant strains were grown overnight at 28°C in XVM2 medium and normalized to an $OD_{600 nm}$ of 0.3 to access the *in vitro* biofilm formation on abiotic surface. The glass tubes were incubated for 24 and 48 h at 28°C without agitation. The planktonic cells were carefully aspirated, and the optical density was measured at 600 nm to access the growth of both strains using spectrophotometer (GENESYS 10S UV-Vis, Thermo Fisher Scientific). The loosely adhered cells were removed by washing the tubes carefully with MQ three times. The sessile cells were stained with 3 mL of 0.1% crystal violet solution. The tubes were incubated for 30 minutes without shaking at room temperature. After incubation, the crystal violet stain was removed carefully, and excess stain was washed off with MQ. For quantification of biofilm, the stain was solubilized

using a solution of 40% methanol and 10% glacial acetic acid. Absorbance was measured at $OD_{570\ nm}$. Eight biological replicates of each strain were performed.

## EPS assay

The ability of wild type and the mutant strain to produce EPS was estimated as previously described (129). Briefly, overnight culture of the wild type and *tssM* mutant strains was normalized to an $OD_{600\ nm}$ of 0.3 in XVM2 media and grown at 28°C with agitation for 48 hours. The bacterial cells were pelleted at 6,000 × *g* for 10 minutes. The supernatant was carefully transferred to a fresh tube. Two volumes of ice-cold 96% ethanol (vol/vol) was added to the supernatant to precipitate EPS; the tubes were incubated at 4°C overnight. After precipitation, the samples were centrifuged at 6,000 × *g* for 30 minutes at 4°C. The supernatant was carefully discarded, and the pellet was air-dried at room temperature overnight. Dried pellets were weighed to quantify EPS produced by the wild type and mutant strains. Three biological replicates of each strain were performed.

## ACKNOWLEDGMENTS

We thank David Young and the staff at Alabama Supercomputer authority and Auburn Easley computing cluster for providing the computational resources necessary for this work.

This work was supported by the Alabama Ag Experiment Station at Auburn University and the National Science Foundation, Division of Integrative Organismal Systems (IOS-1942956) project to Neha Potnis.

## AUTHOR AFFILIATIONS

[1]Department of Entomology and Plant Pathology, Auburn University, Auburn, Alabama, USA
[2]Department of Plant Pathology, University of Florida, Gainesville, Florida, USA

## AUTHOR ORCIDs

Sivakumar Ramamoorthy http://orcid.org/0000-0002-8359-1701
Mathews Paret http://orcid.org/0000-0002-2520-0418
Jeffrey B. Jones http://orcid.org/0000-0003-0061-470X
Neha Potnis http://orcid.org/0000-0002-5418-5894

## FUNDING

| Funder | Grant(s) | Author(s) |
| --- | --- | --- |
| National Science Foundation (NSF) | IOS-1942956 | Neha Potnis |

## AUTHOR CONTRIBUTIONS

Sivakumar Ramamoorthy, Data curation, Formal analysis, Investigation, Methodology, Validation, Visualization, Writing – original draft, Writing – review and editing | Michelle Pena, Methodology | Palash Ghosh, Investigation, Methodology, Visualization, Writing – review and editing | Ying-Yu Liao, Investigation, Methodology | Mathews Paret, Project administration, Writing – review and editing | Jeffrey B. Jones, Conceptualization, Project administration, Writing – review and editing | Neha Potnis, Conceptualization, Data curation, Formal analysis, Funding acquisition, Investigation, Methodology, Project administration, Resources, Software, Supervision, Validation, Visualization, Writing – original draft, Writing – review and editing

## DATA AVAILABILITY

The RNA-Seq reads used in this study are deposited in the National Center for Biotechnology Information Sequence Read Archive database (Bioproject: PRJNA954348). The scripts, input files, intermediate files, and output files used for interpretation of the analyses conducted in this work are made available on https://github.com/Potnislab/T6SS_RNASeq_manuscript as well as DOI: 10.5281/zenodo.8144145 (130).

## ADDITIONAL FILES

The following material is available online.

### Supplemental Material

**Supplemental material (Spectrum02852-23-S0001.pdf).** Fig. S1 and Tables S1 to S13.

### Open Peer Review

**PEER REVIEW HISTORY (review-history.pdf).** An accounting of the reviewer comments and feedback.

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
