## [Reviewer comments · Microbiology Spectrum]

Microbiology Spectrum

Transcriptome profiling of type VI secretion system core gene *tssM* mutant of *Xanthomonas perforans* highlights regulators controlling diverse functions ranging from virulence to metabolism

Sivakumar Ramamoorthy, Michelle Pena, Palash Ghosh, Ying-Yu Liao, Mathews Paret, Jeffrey Jones, and Neha Potnis

Corresponding Author(s): Neha Potnis, Auburn University

Review Timeline:

Submission Date:	July 14, 2023
Editorial Decision:	August 8, 2023
Revision Received:	October 19, 2023
Accepted:	October 20, 2023

Editor: Lindsey Burbank

Reviewer(s): The reviewers have opted to remain anonymous.

Transaction Report:

DOI: <https://doi.org/10.1128/spectrum.02852-23>

August 8, 2023

Dr. Neha Potnis
Auburn University
Department of Entomology and Plant Pathology
209 Rouse Life Science Building
Auburn, AL 36849

Re: Spectrum02852-23 (Transcriptome profiling of type VI secretion system core gene *tssM* mutant of *Xanthomonas perforans* highlights regulators controlling diverse functions ranging from virulence to metabolism)

Dear Dr. Neha Potnis:

Thank you for submitting your manuscript to Microbiology Spectrum. The reviewers agree that your study is well done, but have some suggestions for clarification and improvement. Please note that novelty is not part of the required criteria for acceptance at Microbiology Spectrum, so you can focus your attention on improvements to the current manuscript and ensuring that the conclusions as is are supported by the data. Also note that there are additional reviewer comments in the attached files. When submitting the revised version of your paper, please provide (1) point-by-point responses to the issues raised by the reviewers as file type "Response to Reviewers," not in your cover letter, and (2) a PDF file that indicates the changes from the original submission (by highlighting or underlining the changes) as file type "Marked Up Manuscript - For Review Only". Please use this link to submit your revised manuscript - we strongly recommend that you submit your paper within the next 60 days or reach out to me. Detailed instructions on submitting your revised paper are below.

Link Not Available

Sincerely,

Lindsey Burbank

Journals Department
Reviewer comments:

Reviewer #1 (Comments for the Author):

The work performed by Ramamoorthy and colleagues investigates the global transcriptional changes in *Xanthomonas perforans* driven by deletion of a type-6 secretion system gene, *tssM*. The authors carried out a comparative transcriptomic analysis and used phenotypic experiments (motility, biofilm and EPS levels) to validate the changes in the expression of a few genes. Overall, the manuscript is clearly written, the procedures are properly described, and the results are nicely presented and

interpreted. However, despite interesting data showing that the absence of tssM globally impacts gene expression, the manuscript fails to provide sufficient novelty. An investigation of the molecular basis for the functional connection between T6SS and any of the molecular processes affected by tssM deletion or transcriptomic analyses of cells lacking regulatory genes identified to be upregulated in the tssM, organizing the DEGs into groups of genes dependent on each regulatory gene, could add to the manuscript. Below are suggestions for the current version of the manuscript:

Major points:

1. The Results and Discussion sections are very detailed and redundant descriptions of the transcriptional changes observed upon comparing the strains. By adding more data, the description could be shortened to make the text more friendly.
2. The motility experiment was performed in triplicate, so a plot with mean, error and statistical analysis may be included in addition to the images.
3. RT-qPCR could be used to validate more genes.

Minor points:

1. Indicate in Fig. 1A the time point (8 and 16h) at the top of the corresponding plot.
2. Fig. 2A, 3A and 4. Replace gene number with gene name if possible. This would help the reader interpreting the data, as the text cites gene name, but the figures show gene number in the current version of the manuscript.

Reviewer #2 (Public repository details (Required)):

They have RNAseq results that should be deposited in a public repository

Reviewer #2 (Comments for the Author):

The manuscript by Ramamoorthy et al evaluates the transcriptional effects of the mutation in TssM in *Xanthomonas perforans*. They conclude that a crosstalk exists between the regulation of T6SS and that of other secretion systems, especially the T3SS, a major virulence factor in *Xanthomonads*. The authors perform a thorough analysis of the transcriptional consequences of knocking out this gene, since they report on the differential expression of several functional groups of genes, such as cell wall biosynthesis, stress responses, regulation of gene expression, biofilm formation, phosphorous metabolism. This article provides an opening to the interaction between the T6SS and other systems in the bacterium, something that was suspected before, because of the pleiotropic effects of diverse T6SS mutants, but that was not molecularly tested. They identify some of the candidate genes responsible for this pleiotropic effect and the results that suggest crosstalk between virulence or general signaling systems and T6SS are valuable.

In general, the manuscript is well written and organized in a logical manner, and the conclusions are well supported by the results. However, the major drawback of the manuscript is its extremely descriptive nature. There are a couple of biological supports to the RNAseq analysis, such as the tests for biofilm formation and motility, but the manuscript does not embark into explaining the discrepancies between phenotypes of T6SS mutants reported in the recent literature. Nor does it attempt to explain why this mutant would be hypervirulent in planta or how the T6SS could contribute to the mentioned microbiome-mediated protection, which are central ideas at the beginning of the document. I believe that the manuscript could use more experiments that connect the expression data with the biological relevance of it in pathogenesis.

One major additional concern is experimental: Did the authors measure in vitro growth for the mutant and the wild type? If they have different growth rates, is it at all possible that the transcriptional differences observed are due more to the growth stage than to the mutation of the TssM gene itself?

Most minor concerns are annotated as comments in the PDF, except:

- The authors should explain more deeply what is known about the role of TssM in the bacterium in the introduction
- Please make sure that what the authors describe in the text in the results section corresponds 100% with the results shown in the table, because there are several instances where this was not true. One example is pilT, where the authors report that at 16h has an increase in expression, Table S4 reports increase but not significant difference at 8h.
- Some genes do not appear in the supplementary table
- There is a lack of statistical support in the biological tests. Could the authors please include these analyses?
- Please make sure that all references that are needed are cited. Sometimes there is no citation of a reference in important parts of the manuscript

Staff Comments:

Preparing Revision Guidelines

Please return the manuscript within 60 days; if you cannot complete the modification within this time period, please contact me. If you do not wish to modify the manuscript and prefer to submit it to another journal, please notify me of your decision immediately so that the manuscript may be formally withdrawn from consideration by Microbiology Spectrum.

Transcriptome profiling of type VI secretion system core gene *tssM* mutant of *Xanthomonas perforans* highlights regulators controlling diverse functions ranging from virulence to metabolism

Sivakumar Ramamoorthy¹, Michelle Pena¹, Palash Ghosh¹, Ying-Yu Liao², Mathews Paret², Jeffrey B Jones² and Neha Potnis^{1*}

¹ Department of Entomology and Plant Pathology, Auburn University, AL 36849

² Department of Plant Pathology, University of Florida, Gainesville, FL 32611

*Corresponding author: email: nzp0024@auburn.edu

Running title: T6SS regulation in *Xanthomonas*

Abstract

Type VI secretion system (T6SS) is a versatile, contact dependent contractile nano-weapon in Gram-negative bacteria that fires proteinaceous effector molecules directly into prokaryotic and eukaryotic cells aiding in manipulation of the host and killing of competitors in complex niches. In plant pathogenic xanthomonads, T6SS has been demonstrated to play these diverse roles in individual pathosystems. However, the regulatory circuit involved in mediating biological functions carried out by T6SS are still elusive in *Xanthomonas* sp. To bridge this knowledge gap, we conducted an *in vitro* transcriptome screen using plant apoplast mimicking minimal medium, XVM2 medium, to decipher the effect of *tssM* deletion, a core gene belonging to T6SS-cluster i3*, on the regulation of gene expression in *Xanthomonas perforans* strain AL65. Transcriptomic data revealed that a total of 277 and 525 genes were upregulated, while 307 and 392 genes were downregulated in the mutant strain post 8 and 16 hours of growth in XVM2 medium. The transcript abundance of several genes associated with flagellum and pilus biogenesis as well as type III secretion system were downregulated in the mutant strain. Deletion of *tssM* of cluster-i3* resulted in upregulation of several T6SS genes belonging to cluster-i3*** and genes involved in biofilm and cell wall biogenesis. Similarly, transcription regulators like *rpoN*, Pho regulon, *rpoE* and *csrA* were identified to be upregulated in the mutant strain. Our results suggest that T6SS modulates the expression of global regulators like *csrA*, *rpoN* and *pho* regulons triggering a signaling cascade and coordinates the expression of suite of virulence factors, stress response genes and metabolic genes.

Keywords: Plant pathogen interaction, secretion system, T6SS, transcriptome sequencing, T3SS, biofilm.

Importance

Type VI secretion system (T6SS) has received attention due to its significance in mediating inter-organismal competition through contact-dependent release of effector molecules into prokaryotic and eukaryotic cells. Reverse-genetic studies targeting core genes of T6SS have indicated the role of T6SS in virulence in a variety of plant pathogenic bacteria, including *Xanthomonas* studied here. However, it is not clear whether such effect on virulence is merely because of a shift in the microbiome-mediated protection or if T6SS is involved in a complex regulatory network governing virulence in plant pathogens involving type III secretion system or c-di-GMP signaling pathways. In this study, we conducted *in vitro* transcriptome profiling in minimal medium to decipher the signaling pathways regulated by *tssM-i3** in *Xanthomonas perforans* strain AL65. We show that TssM-i3* regulates the expression of a suite of genes associated with virulence and metabolism either directly or indirectly by altering the transcription of several regulators. These findings further expand our knowledge on the intricate molecular circuits regulated by T6SS in phytopathogenic bacteria.

Introduction

Bacterial secretion systems are characterized into nine different groups (Tat system, types I–VII, type IX) according to their structure, function, and effector molecules that they deliver (1–4). Of these different secretion systems, the type VI secretion system (T6SS) is present in nearly 25% of Gram-negative bacteria with at least one T6SS encoded in their genome. T6SS is known to perform a wide range of functions in host-associated commensals, pathogens as well as free-living bacteria (5–14). The T6SS is a needle-like contact dependent molecular nanomachine that injects toxic effectors directly into cognate eukaryotic and prokaryotic cells (15). The delivered effectors subvert or manipulate host cells or fight against bacterial competitors residing in complex environmental niches (15, 16). The assembly, delivery of effectors and disassembly of T6SS is highly dynamic. Upon induction by an unknown signal, the T6SS is assembled as a puncturing device comprising 13 essential core components, with three major proteins, TssJ, TssL and TssM constituting the membrane complex anchored to the cytoplasmic baseplate structures. The effectors can interact specifically with V

proteins (17–19), the piercing device of the T6SS apparatus, or can form an effector-VgrG complex resulting in an evolved VgrG (20, 21) or conjugate with PAAR domain which packs the toxins onto the associated VgrG (22, 23). This process is tightly synchronized by complex signaling cascade events.

Regulation of T6SS happens at different levels: transcriptional, post-transcriptional and post-translational. Transcriptional control of T6SS genes is highly diverse among bacterial species and numerous components including sigma factor *rpoN* (σ^{54}) (24, 25), transcriptional regulator FleQ (26), nucleoid structuring protein H-NS (27), nutrient acquisition pathway regulators such as Fur (28), PhoB-PhoR, and secondary messenger cyclic diguanylate (c-di-GMP) have been known to regulate the activity of T6SS (26, 29, 30). Quorum sensing (QS) plays a vital role in regulating the expression of genes implicated in T6SS. For example, the T6SS was reported to be regulated by QS system in numerous bacteria, such as *Vibrio cholerae* (31), *Vibrio fluvialis* (32), *Pseudomonas aeruginosa* (33), *Burkholderia thailandensis* (34) and *Aeromonas hydrophila* (35). Post-transcriptional regulation of T6SS genes is well characterized in *Pseudomonas* spp. The Gac/Rsm regulatory cascade controls T6SS by modulating the expression of a RNA binding protein, RsmA, which represses T6SS activity (36, 37). Two separate models have been proposed for post-translational control of T6SS gene cluster: phosphorylation-dependent and phosphorylation-independent pathway (38, 39). Since the discovery of T6SS, key breakthroughs are described in animal pathogens elucidating several mechanisms, functional attributes and environmental cues activating T6SS (40–46). The expression of T6SS can be regulated by several environmental signals like metal ion scarcity, reactive oxygen species, pH, temperature, osmotic stress, and membrane damage (28, 47–49) but little is known on the regulation of T6SS, and traits regulated by T6SS in phytopathogens.

In phytopathogenic bacteria, maintenance of a single or simultaneous multiple clusters of T6SS, their non-random distribution across phylogenetic clades and gene flow within genera suggest the importance of this energy-expensive machinery in pathogen biology (5, 50). For example, *X. perforans* contains two T6SS clusters, i3* and i3*** (50, 51). In recent reverse genetics studies, deletion mutagenesis of core T6SS genes in different phytopathogenic bacteria revealed pleiotropic effects of T6SS on various phenotypes such as biofilm formation, motility, competition with other prokaryotic or eukaryotic members, and virulence (10, 52, 53). Interestingly, inactivation of T6SS does not always lead to reduction in virulence

even when comparing within genera. For example, mutation in *hcp* of T6SS-2 of *Xanthomonas oryzae* led to a reduction of virulence in rice, whereas our previous work indicated that mutation in *tsi3** of T6SS i3* of *X. perforans* led to higher aggressiveness with faster symptom development during pathogenesis (51). Similar observations have been made suggesting variable contribution of T6SS towards virulence in other genera including *P. syringae* (8, 54), *A. tumefaciens* (9, 55) and *P. ananatis*. Such varied contributions of T6SS in virulence in different pathosystems suggest that the role in T6SS regulation may be different depending on the pathosystem or mode of infection. In pathogens where direct impact of T6SS on virulence has been validated, this may indicate involvement of T6SS in regulation of known virulence factors, such as T3SS and effectors. The differences in contribution towards virulence of T6SS mutants can also be attributed to the method of inoculation, suggesting the importance of spatial regulation of T6SS. For example, *X. euvesicatoria vgrG* and *clpV* mutants that showed no difference in virulence when compared to wild type upon infiltration directly into apoplast (56), whereas T6SS core gene mutant of closely related sister species, *X. perforans*, showed hypervirulence on tomato upon dip-inoculation (51). This work also revealed the contribution of T6SS-i3* towards osmotolerance and higher epiphytic survival in competition experiment, suggesting epiphytic fitness imparted by functional T6SS-i3* in this foliar hemibiotrophic pathogen. Similar findings highlighting the induction of T6SS during epiphytic colonization of citrus in *X. citri* but low expression in apoplast suggest spatio-temporal regulation of T6SS during infection (57). In the foliar hemibiotroph, *P. syringae*, upregulation of genes associated with T6SS was observed in cells exposed to increased osmotic stress, a stress commonly encountered in the phyllosphere environment (58). Another stress that pathogens experience during their colonization in the hostile phyllosphere environment is competition with the resident flora. In *Xanthomonas citri*, resistance to predation by the soil amoeba *Dictyostelium* was mediated by T6SS (11). In addition, the T6SS is also known to be involved in niche modification by secreting metal-scavenging proteins (59). Until now, the role played by T6SS in fitness of plant pathogens during host colonization has relied on derived phenotypic data of T6SS mutants, however the mechanistic basis of these observations and potential direct or indirect effects on known virulence factors have not been completely deciphered. We do not know the cues involved in the activation of T6SS in phytopathogenic bacteria, whether they are limited to competitors or certain plant derived environmental signals. Assuming that T6SS is induced in response to both of these cues in foliar pathogens, it is possible that gene expression patterns might have little or no overlap when comparing across individual cues. For example, T6SS might be

involved in regulating genes involved in observed phenotypes associated with overall epiphytic fitness such as osmotolerance, overcoming the nutrient limitation, or competition with resident flora during early colonization events of the pathogen. However, as pathogen switches from epiphytic to apoplastic colonization, T6SS might be involved in regulating major virulence genes for successful infection. Thus, such spatio-temporal regulation of virulence during epiphytic or apoplastic pathogen infection makes it harder to tease apart similarities or differences in gene expression patterns in response to plant cues or microbiota cues and explain underlying phenotypes using *in planta* transcriptome screen (60). In contrast, *in vitro* system allows to discern these differences or similarities using controlled condition experiments. Minimal medium or nutrient deprived synthetic medium that mimic plant apoplastic environment such as XVM2, XOM2 and XOM3 have been used previously as an *in vitro* system to trigger the virulence factors of phytopathogen (61–63). Thus, we conducted an *in vitro* transcriptome screen to understand the genome-wide expression patterns of T6SS-i3* core gene *tssM* mutant of foliar tomato/pepper pathogen, *X. perforans*, (also known as *X. vesicatoria* pv. *perforans*), under apoplast-mimicking minimal medium. Our results provide insight into the regulatory pathways modulated by T6SS which will be valuable to better understand the pathogenesis of this important phytopathogen.

Results

Transcriptome analysis and COG based gene expression pattern.

To gain insights into the molecular mechanisms regulated by *tssM* in AL65, a genome-wide *in vitro* transcriptome analysis was performed by culturing the wild type and the mutant strains in XVM2 medium. Analysis of RNA-Seq data revealed that in the mutant strain, a total of 277 and 525 genes were upregulated relative to the wild type strain, while 307 and 392 genes were downregulated post 8 and 16 h of growth in XVM2, respectively (Fig. 1A). The complete list of DEGs is provided in Table S1. Among the differentially expressed genes, 163 genes were upregulated at both 8 and 16 h and 114 genes were downregulated at both 8 and 16 h. Interestingly, 23 genes downregulated at 8 hours were found to be upregulated at 16 hours (Fig. 1B) (Table S2). Functional categorization of DEGs in the mutant strain showed genes associated with motility and signal transduction were highly downregulated, whereas genes implicated with ribosomal protein synthesis, cell membrane biogenesis and intracellular trafficking and secretion were upregulated (Fig. 1C).

Figure 1. *In vitro* transcriptome profiling revealed nearly 14% and 22% of genes were differentially expressed in the *tssM* mutant post 8 and 16 h of growth in XVM2 medium. (A) Volcano plot illustrating the upregulated and downregulated genes in *tssM* mutant at 8 h (left) and 16 h (right). Each dot indicates an individual coding sequence with detectable expression in the mutant strain in comparison to the wild type. The X-axis represents the \log_2 fold change between mutant and wild type, and the Y-axis represents the $-\log_{10}$ transformed adjusted *p*-value denoting the significance of differential expression. (B) Venn diagram of genes with significant difference in transcript level between mutant and wild type. The genes showing overlap between the two time points are represented in percentage. (C) Functional categorisation of differential expressed genes in the *tssM* mutant based on the clusters of orthologous groups (COGs).

Deletion of *tssM* gene resulted in downregulation of genes associated with motility and chemotaxis

We identified many genes associated with motility such as *fliDE*, *HIJKNOPS*, *flgABCDEFGHIJKLMN*, *flhFAB* and *motABD* that were significantly downregulated in the mutant strain after *in vitro* growth in XVM2 medium for 8 and 16 h (Table S3). These genes play a vital role in flagella dependent swimming motility. In addition, the master regulator sigma factor FliA was found to be downregulated in the mutant strain at both time points. Further, a motility assay was performed to compare the abilities of the wild type and mutant strains to swim on a semi-solid medium. The mutant strain displayed significantly reduced

swimming halos in comparison to the wild type which was in accordance with this RNA-Seq data (Fig 2).

Figure 2. Deletion of *tssM* resulted in downregulation of flagella biosynthesis genes. (A) Heat map showing expression pattern of genes involved in flagella production and chemotaxis. (B) Swimming motility assay for the wild type and *tssM* mutant. The strains were inoculated into nutrient and XVM2 containing 0.2% agar. The wild type displayed higher swimming halo in comparison to the mutant strain. The experiment was performed in triplicate and the representative colony morphology of the wild type and mutant strains were photographed.

The transcript level of genes involved in type IV pilus biogenesis (*pilABGMNOPQTU*) involved in major pilin and ATPase activity was reduced, while the genes associated with minor pilin (*pilEVWXYI*) were upregulated in the mutant strain at 8 hours (Table S4). The mutant strain showed reduced twitching motility in both nutrient and XVM2 medium (Fig. 3). Interestingly, in addition to the minor pilin, six genes (*pilMNOPTU*) were downregulated at 8 h and part of major pilin were upregulated at 16 h in the mutant strain (Table S4). Several genes involved in chemotaxis like *cheW*, *cheR*, *cheY*, *cheB* and sensory receptor methyl accepting chemotaxis proteins displayed significant downregulation in the mutant strain at 8 and 16 h (Table S3).

Figure 3. The transcript level of type IV pili genes was differentially expressed in the *tssM* mutant. (A) Heat map showing expression of differentially expressed genes related to type IV pili biogenesis. (B) Colony margin characteristics of wild type and mutant strains grown on nutrient and XVM2 containing 1% agar. The wild type exhibited distinct peripheral fringe indicative of type IV pili mediated twitching motility. Magnification scale, 1 mm.

c-di-GMP turnover genes were downregulated in the *tssM* mutant

The genome of *X. perforans* strain AL65 contains 31 genes encoding for GGDEF, EAL and HD-GYP domain containing protein. Among them, the transcript level of 9 genes coding for diguanylate cyclase (DGC) with GGDEF domain, 3 genes encoding for EAL domain containing protein and 1 gene with HD-GYP domain involved in phosphodiesterase (PDE) activity and 4 genes coding for GGDEF/EAL domain containing proteins were found to be significantly downregulated in the mutant strain at 8 h (Table S5) (Fig. 4).

Figure 4. Mutation of *tssM* resulted in dysregulation of genes implicated in various cellular processes. Transcriptional pattern of differentially expressed genes varying from virulence to metabolism. Heat maps showing the intensity of expression of each gene (rows) expressed as normalized read counts and scaled for each row. Each column represents the samples used in the study and is clustered based on the result of hclust analysis.

Downregulation of T3SS core genes, regulators and effectors was observed in the mutant.

The expression of T3SS genes is modulated by *hrpG* and *hrpX*, key master regulators of virulence (64). We observed that genes associated with T3SS were downregulated in the mutant strain at 16 h. The expression of two genes, *hrpX* (transcriptional regulator) and *hrpG* (response regulator), was significantly downregulated by 2.6-fold (Table S6, Fig 4). The

expression of several *hrp* genes that shape the core structural apparatus of T3SS (*hrpB1*, *hrpB2*, *hrpB4*, *hrpB7*, *hrpF*) were downregulated in the mutant strain at 16 h (Table S6) (Fig. 4). The structure of the injectisome begins with the formation of export apparatus and the membrane rings. The RNA-Seq data revealed 10 genes *hrcC*, *hrcJ*, *hrcL*, *hrcN*, *hpaC*, *hrcQ*, *hrcR*, *hrcS*, *hrcT* and *hrcU*, which are implicated in the formation of export apparatus, cytosolic ring, and ATPase activity, were downregulated in the mutant strain at 16 h (Table S6). In addition to discharging virulence factors, T3SS also plays a key role in host cell sensing and modulation of gene expression post attachment to the eukaryotic host. This function is primarily regulated by the T3SS specific chaperone CesT/Hpa3, which was downregulated 2.8-fold in the mutant strain at 16 h. The mRNA levels of four genes coding for type III effector proteins, HpaF, XopK, AvrXv4 and F-box, were significantly decreased between 2.5- and 1.8-fold in the mutant strain at 16 h. Similarly, the transcript levels of two genes (E2P69_RS17705 and E2P69_RS22950) encoding XopP effector paralogs were downregulated by 2.4-fold in the mutant strain at 16 h.

***tssM* negatively regulates genes associated with T6SS under *in vitro* condition**

Two T6SS gene clusters are detected in the genome of *X. perforans* strain AL65, *i3** and *i3****. Surprisingly, our transcriptome data revealed that deletion of *tssM*, a core gene associated with T6SS cluster *i3** resulted in upregulation of T6SS genes forming the *i3** and *i3**** gene clusters in the mutant strain at 16 h (Fig. 4). Ten genes belonging to cluster *i3** were upregulated in the mutant strain at 16 h. Similarly, ten genes belonging to T6SS cluster *i3**** displayed increased expression at 16 h (Table S7). The transcript abundance of a gene coding for OmpA family protein, which is part of T6SS accessory genes, increased in the mutant strain at 16 h. We further identified the expression of seven genes (i.e., E2P69_RS05680, E2P69_RS08700, E2P69_RS17420, E2P69_RS09365, E2P69_RS19505, E2P69_RS00975 and E2P69_RS14695) that code for RHS repeat-associated core domain-containing protein, four tetratricopeptide repeat proteins, phospholipase D family protein and cardiolipin synthase B were significantly upregulated in the mutant strain at 16 h. These proteins constitute the T6SS-dependent effectors facilitating an array of interactions with prokaryotes and eukaryotes. The mRNA level of sensor kinase VirA, part of *virAG* two component system, which is known to positively modulate the expression of T6SS gene cluster in *Burkholderia* (65–67), was significantly upregulated. Post-translational (serine-threonine kinase, E2P69_RS09420; serine-threonine phosphatase, E2P69_RS09415) regulators of T6SS were identified to be upregulated in the mutant strain (Table S7).

Formation of disulfide bonds in secreted effector proteins is crucial for their function and stability. In this RNA-Seq screen, the mRNA levels of two genes (E2P69_RS12845 and E2P69_RS03325) coding for thiol:disulfide interchange protein DsbA/DsbL and thiol-disulfide oxidoreductase DCC family protein displayed increase abundance in the mutant strain by 2.5- and 2-fold respectively.

Genes involved in T2SS, T4SS and T5SS were differentially expressed in the *tssM* mutant

Several degradative enzymes like proteases, lipases, cellulase and xylanases are secreted via T2SS. Genes coding for extracellular enzymes were identified to be significantly dysregulated in the mutant strain post growth in XVM2 medium (Table S8). The mRNA level of three genes (E2P69_RS00220, E2P69_RS00225 and E2P69_RS00235) coding for glycoside hydrolase family 5 protein was significantly downregulated in the mutant strain at 8 and 16 h. Blast analysis of the amino acid sequence showed this protein belongs to BglC superfamily which possess cellulolytic activity. Another gene (E2P69_RS19010) coding for cellulase family glycosyl hydrolase was significantly downregulated by 2-fold in only 8 h. The transcript abundance of two lipase coding genes (E2P69_RS02215 and E2P69_RS15945) was significantly decreased by 1.8- and 1.7- fold in the mutant strain at 16 h. The mRNA level of xylan degradation proteins encoded by E2P69_RS05520 and E2P69_RS19555 was upregulated in the mutant strain at 16 h while one gene (E2P69_RS05380) coding for endo-1,4-beta-xylanase was downregulated in the mutant strain at 16 h. The transcript level of one gene (E2P69_RS15315) coding for alpha-amylase protein family showed an increase in expression in the mutant strain at 16 h. In *Xanthomonas* spp. type V secretion system (T5SS) is divided into three subgroups (Va, Vb and Vc). The expression of two genes (E2P69_RS19840 and E2P69_RS22570) coding for autotransporter domain-containing esterase and YadA-like family protein was identified to be upregulated in the mutant strain at 8 and 16 h respectively. These genes belong to the subgroup Va and Vc while one gene (*phaB*) under Vb subgroup and two periplasmic chaperons (VirK and DegQ) were downregulated in the mutant strain. The degradative enzymes and T5SS proteins are translocated across the bacterial membrane by the Sec or TAT (twin arginine transport) pathway. Our RNA-Seq analysis revealed two Sec pathway genes (E2P69_RS22135 and E2P69_RS22465) were upregulated in the mutant strain at both time points while two genes (E2P69_RS05570 and E2P69_RS05575) belonging to TAT pathway were significantly downregulated in the mutant at 16 h. We identified five genes (E2P69_RS14500,

E2P69_RS19420, E2P69_RS14505, E2P69_RS14525, E2P69_RS19425) associated with type IV secretion system (T4SS) in the mutant strain at 16 h that were downregulated at 8 h, while five genes (E2P69_RS21420, E2P69_RS21425, E2P69_RS21460, E2P69_RS21450, E2P69_RS21445) associated with the T4SS that were upregulated in the mutant strain at 16 h (Table S8).

Expression of genes involved in transcription, posttranscriptional, and translational control of gene expression is modulated by *tssM*.

The transcriptome data revealed that deletion of *tssM* resulted in differential expression of several genes associated with transcription, post-transcription, and translational control of gene expression. The alternative sigma factor coding for RNA polymerase factor sigma-54 (RpoN or σ^{54}) was upregulated in the mutant strain at 16 h which plays a role in balancing intracellular carbon and nitrogen levels. Similarly, a global regulator (*csrA*, E2P69_RS21330) associated with post-transcriptional regulation of gene expression was upregulated in the mutant strain at both 8 and 16 h (Table S9). RNase plays a vital role in posttranscriptional modulation of gene expression in pathogenic bacteria aiding the organism to swiftly adapt to the host environment. In the transcriptome analysis we identified the mRNA level of two genes (E2P69_RS00065 and E2P69_RS10975) coding for ribonuclease P protein component and ribonuclease HII was significantly upregulated in the mutant strain at 8 and 16 h while two genes (E2P69_RS10175 and E2P69_RS09245) coding for Ribonuclease PH and rRNA maturation RNase YbeY was significantly downregulated in the mutant strain at 8h.

Several genes coding for ribosomal subunit protein were identified to be highly upregulated in the mutant strain at both time points (Table S9) (Fig. 4). The ribosomal proteins constituted for 26% and 12% of upregulated genes in the mutant strain at 8 and 16h post growth in XVM2 medium. The composition of bacterial ribosomal proteins is highly heterogenous and has paralogs of some ribosomal proteins that have evolved to combat environmental stress. For example, the transcript abundance of two genes (E2P69_RS10215 and E2P69_RS08540) coding for type B 50S ribosomal protein L31 and L36 was significantly higher in the mutant strain at 8 and 16 h. These genes play a vital role in alleviating zinc starvation stress (68). In addition to genes involved in ribosomal protein biogenesis, the transcript levels of several genes implicated in maturation and assembly of ribosomes were significantly increased in the mutant strain. Ribosome assembly factors like GTPase (ObgE, Der, HflX), maturation proteins (RimM and RimP) and protein chaperone

(RbfA) were upregulated at both time points, whereas the transcript abundance of one gene (E2P69_RS07135) coding for RNA binding protein YhbY associated with ribosome assembly was decreased in the mutant strain at 8 h. The precision and efficiency of protein synthesis can be modulated by tRNA modification. Queuosine tRNA modification is widespread in prokaryotes which play a role in efficacy and accuracy of translation. The mRNA level of a gene (E2P69_RS04970) involved in queuosine metabolism was significantly upregulated 2.2-fold in the mutant strain at 8 h. One gene coding for ribosome-associated translation inhibitor RaiA was identified to be significantly downregulated in the mutant strain at 16 h. This gene plays an important role in ribosome hibernation resulting in loss of translational activity of the ribosomes. Indeed, this downregulation of translation inhibitor aligns with the observation of significant upregulation of various ribosomal subunits encoding genes in the mutant.

Deletion of *tssM* resulted in upregulation of genes involved in cell wall biogenesis

Several genes coding for cell wall biogenesis were found to be significantly upregulated in the mutant strain. Nine genes coding for glycosyltransferases (E2P69_RS06005, E2P69_RS19735, E2P69_RS06015, E2P69_RS05035, E2P69_RS19740, E2P69_RS05995, E2P69_RS05990, E2P69_RS17085, E2P69_RS07810) were upregulated at 16 h (Table S10). These enzymes are essential for the formation of repeating units of monosaccharide moieties which are the basic units for biosynthesis of peptidoglycan, exopolysaccharides (EPS) and lipopolysaccharides (LPS). Genes involved in the biosynthesis of LPS displayed significantly increased expression in the mutant strain. Two genes (E2P69_RS10970 and E2P69_RS03820) coding for O-antigen ligase family protein and lipid-A-disaccharide synthase were found to be upregulated at 8 and 16 h, while one gene (E2P69_RS05050) involved in O-antigen modification was upregulated only at 16 h. The transcript level of three genes (E2P69_RS22825, E2P69_RS08060, E2P69_RS08055) coding for tetraacyldisaccharide 4'-kinase LpxK and mannose-1-phosphate guanylyltransferase/mannose-6-phosphate isomerase, 3-deoxy-manno-octulosonate cytidyltransferase was upregulated at 16 h. These genes are crucial players in the biosynthesis of LPS. Four genes associated with EPS production coding for glycosyltransferase (E2P69_RS06005), acyltransferase family (E2P69_RS05980), *gum* cluster cupin domain (E2P69_RS06020) and *gumC* family protein (E2P69_RS05965) part of *gum* cluster were identified to be significantly upregulated in the mutant strain at 8 h. In-vitro assays were conducted to quantify the biofilm and exopolysaccharide production ability of wild type and mutant

strains. There was no significant difference in the biofilm forming ability between the wild type and the mutant strains. In-vivo assay revealed increased EPS present in the *tssM* mutant at 48 h although the difference was not significant (Fig. S1). Four genes encoding for UDP-3-O-(3-hydroxymyristoyl) glucosamine N-acyltransferase, acyl-ACP-UDP-N-acetylglucosamine O-acyltransferase, polysaccharide pyruvyl transferase family protein and murein biosynthesis integral membrane protein MurJ were found to be upregulated in the mutant strain at 16 h (Table S10) (Fig. 4). These proteins play a major role in biosynthesis of peptidoglycan. In addition, genes implicated in remodeling of cell walls for anchoring surface appendages like flagella, pili, and channels for translocation of DNA, nutrients and protein effectors across membrane were also found to be differentially expressed in the mutant strain. A transglycosylase (E2P69_RS14115) encoding gene was upregulated in the mutant strain at 8 and 16 h, while a gene coding for lytic murein transglycosylase MltB (E2P69_RS14605) was upregulated only at 16 h. The expression of two genes (lytic transglycosylase domain-containing protein MltE and flagellar assembly peptidoglycan hydrolase FlgJ) were downregulated in the mutant strain at 8 and 16 h. The latter gene *flgJ* is essential for the assembly of flagellar apparatus on the bacterial cell membrane.

The expression of metabolic and inorganic ion transport genes is differentially regulated by *tssM* in AL65 under *in vitro* condition

Several genes involved in nitrogen metabolism were significantly upregulated in the mutant strain. The expression of other nitrogen-regulated genes under *rpoN* regulon like *narA*, *nirBD*, *ntnC*, *glnK-a*III were significantly increased in the mutant strain (Table S11). Like nitrogen metabolism, several genes implicated in phosphate (Pi) homeostasis were found to be significantly upregulated in the mutant strain. One gene coding for sensor histidine kinase PhoR, part of two component system (TCS) PhoBR was upregulated in the mutant strain at 16 h. The transcription of genes under Pho regulon is modulated by PhoBR in response to external Pi concentration. Similarly, the transcript abundance of a gene coding of PhoH family protein belonging to the core member of Pho regulon was significantly increased by 1.56-fold in the mutant strain at 16 h. In addition to the TCS PhoBR, Pi homeostasis is also maintained by the Pst system. In our RNA-seq analysis, all four genes of Pst operon (*pstSCAB*) were found to be significantly upregulated in the mutant strain at 16 h. The phosphate scavengers like phytase (E2P69_RS20335), alkaline phosphatase (E2P69_RS18060) and glycerophosphodiester phosphodiesterase (E2P69_RS16410) were upregulated in the mutant strain at 16 h. Four genes (E2P69_RS02800, E2P69_RS02810,

E2P69_RS02805 and E2P69_RS02815) coding for pyrroloquinoline quinone biosynthesis protein PqqB, pyrroloquinoline quinone biosynthesis peptide chaperone PqqD, pyrroloquinoline-quinone synthase PqqC and pyrroloquinoline quinone biosynthesis protein PqqE involved in phosphate solubilization was downregulated in the mutant strain at 16 h. One gene coding for a phosphate specific porin (E2P69_RS15410) was downregulated in the mutant strain at 8 h. Several genes involved in sulfur transport and metabolisms were identified to be differentially expressed in the mutant strain. Three genes (E2P69_RS07775, E2P69_RS12595 and E2P69_RS07790) associated with sulfur uptake were upregulated at 8 and 16 h while four genes (E2P69_RS13380, E2P69_RS07925, E2P69_RS20745 and E2P69_RS20680) coding for TauD/TfdA family dioxygenase, sulfurtransferase, cysteine synthase A and sulfate adenylyltransferase subunit CysD were downregulated in the mutant strain at 16 h.

Transcriptome analysis revealed differential expression of several metabolic genes in the mutant strain. The transcript level of 10 genes (*hisA-B-C-D-F-G-H-IE*) involved in histidine biogenesis was significantly upregulated in the mutant strain at 16 h (Table S12). De Novo synthesis of histidine plays an essential role in cellular metabolism. Similarly, five genes (E2P69_RS08790, E2P69_RS08775, E2P69_RS08795, E2P69_RS08780 and E2P69_RS21025) associated with arginine biosynthesis pathway were significantly upregulated in the mutant strain at 16 h. Arginine biosynthesis was considered a virulence trait as demonstrated by pathogenesis studies in Xanthomonadaceae species (Sagawa et al., 2022). Three genes (E2P69_RS07515, E2P69_RS19925 and E2P69_RS16500) coding for 3-phosphoshikimate 1-carboxyvinyltransferase, 3-deoxy-7-phosphoheptulonate synthase and aminodeoxychorismate synthase component I was upregulated in the mutant strain at 8 and 16 h, while one gene (E2P69_RS04815), which encodes for chorismate mutase, was upregulated at 8 h in the mutant strain. These genes are important for aromatic amino acid biosynthesis. The transcript abundance of genes associated with leucine biosynthesis pathway was significantly downregulated in the mutant strain. Two genes (E2P69_RS09875 and E2P69_RS09870) coding for 3-isopropylmalate dehydrogenase and 3-isopropylmalate dehydratase small subunit was downregulated at 16 h while one (E2P69_RS09865) encoding for 3-isopropylmalate dehydratase large subunit was downregulated in the mutant strain at 8 and 16 h (Table S12). In addition to amino acid metabolism, the mRNA levels of genes involved in carbohydrate transport and metabolism were identified to be differentially expressed. Six genes (E2P69_RS08130, E2P69_RS08125, E2P69_RS08120 and

E2P69_RS08110, E2P69_RS08725 and E2P69_RS08195) encoding for succinate dehydrogenase cytochrome b556 subunit, succinate dehydrogenase hydrophobic membrane anchor protein, succinate dehydrogenase flavoprotein subunit, succinate dehydrogenase iron-sulfur subunit, cytochrome ubiquinol oxidase subunit I and c-type cytochrome were downregulated in the mutant strain at 8 and 16 h, whereas the transcript level of genes coding for cytochrome c maturation and cytochrome b biogenesis was upregulated in the mutant strain. These genes play an important role in electron transport chain and energy production (Table S12). The transcript level of two genes (E2P69_RS13540 and E2P69_RS21585) encoding for protocatechuate 3,4-dioxygenase subunit alpha and protocatechuate 3,4-dioxygenase subunit beta was lower at 16 h in the mutant strain. These proteins play an essential role in degradation of 4-hydroxybenzoate, an intermediate used in xanthomonadin pigment synthesis (69).

Deletion of *tssM* results in upregulation of genes involved in stress response.

Two genes (E2P69_RS16345 and E2P69_RS16335) involved in α -glucan biogenesis were upregulated in the mutant strain at 16 h (Table S13). Similarly, two genes (E2P69_RS16330 and E2P69_RS16340) responsible for trehalose biosynthesis were upregulated in the mutant strain at 16 h while one gene coding for alpha, alpha trehalase TreA, involved in degradation of trehalose was found to be downregulated in the mutant strain at 8 h. These genes are essential for alleviating abiotic stress. Polyamines play an important role in DNA replication, RNA translation and cell membrane integrity by binding to polyanionic compounds. The major polyamines found in bacteria are putrescine and spermidine. Our RNA-seq analysis revealed six genes (E2P69_RS08795, E2P69_RS12820, E2P69_RS15510, E2P69_RS12825, E2P69_RS08555) involved in synthesis of putrescine and spermidine were upregulated in the mutant strain at 16 h. Three genes (E2P69_RS09280, E2P69_RS09275, E2P69_RS06565) coding for ABC transporter permease PotA, PotB and PotC involved in exchange of polyamines were found to be upregulated in the mutant strain. Similarly, two genes (E2P69_RS14465 and E2P69_RS11830) coding for excinuclease ABC subunit UvrB and excinuclease ABC subunit UvrA was upregulated in the mutant strain. The mRNA level of a general stress protein CsbD (E2P69_RS15800) was significantly increased by 2.2-fold in the mutant strain at 8 h. The transcript abundance of two genes (E2P69_RS10375 and E2P69_RS01815) coding for OsmC family protein and OmpW family protein were downregulated in the mutant strain at 16 h by 1.8- and 1.5- fold respectively. Interestingly, we identified two genes (E2P69_RS15710 and E2P69_RS09345) coding for sigma factors

RpoE that were upregulated in the mutant strain at both time points. This sigma factor plays a crucial role in stress response in bacteria (Table S13).

Discussion

The type VI secretion system (T6SS) is a multifunctional protein firing apparatus in Gram-negative bacteria. Although genes associated with T6SS have been identified in many species of *Xanthomonas*, an important plant pathogen, the biological and molecular functions are not completely understood in many xanthomonads (14). Previously, we reported deletion of *tssM*, a core T6SS *i3** gene, resulted in hypervirulent phenotype of *X. perforans* strain AL65 (51). To characterize the regulatory circuit modulated by TssM, we performed a transcriptome analysis by growing the wild type and the mutant strain in *hrp* inducing medium, XVM2 medium. Based on our findings, TssM of T6SS-*i3** directly or indirectly controls the transcription level of several genes that participate in bacterial virulence (motility, secretion system and biofilm), stress response and metabolic pathway (amino acid and carbohydrate metabolism) through a regulatory network (Fig. 5), indicating its pleiotropic effects.

In the *tssM* mutant, transcript level of flagellar biogenesis genes was downregulated revealing that functional TssM positively regulated these determinants under the tested condition. Flagella is essential for motility and chemotaxis of bacterial cells. The contribution of flagella in virulence and pathogen-associated molecular pattern was established in numerous pathogenic bacteria, including *X. campestris* pv. *campestris* (70–73). Similarly, TssM positively regulates the expression of c-di-GMP turnover genes. In this study, several genes involved in synthesis and degradation of c-di-GMP were identified to be downregulated in the mutant. Since genes coding for both GGDEF and EAL/HD-GYP domain containing proteins were downregulated in the mutant strain, further quantification of the intracellular levels of c-di-GMP in the mutant and the wild type strains would help to understand the dynamic mechanistic framework underlying regulation of virulence factors by T6SS via c-di-GMP.

Figure 5. Model illustrating the regulatory cascade modulated by *tssM* in AL65 under *in vitro* condition. The proposed model provides an overview into the molecular regulation of different functions in *tssM* mutant ranging from virulence to metabolism controlled by major regulators identified in the present study. One major regulator of virulence factors is the nucleotide secondary messenger c-di-GMP. The intracellular levels of c-di-GMP negatively regulate flagella and T3SS (74, 75) while having a positive effect on biofilm formation. The higher transcript level of *gum* cluster genes positively regulates the production of exopolysaccharide. In *Xanthomonas* sp. the secretion and delivery of toxic effectors via T3SS is controlled by two major transcriptional regulators HrpG and HrpX (76). Phosphorylated HrpG binds directly to the promoter region of *hrpX*. HrpX induces the expression of T3SS and T3Es coding genes by binding to the plant inducible promoter (PIP) box, a DNA motif found in the structural T3SS operon and TSEs gene. The RNA binding protein, CsrA positively regulate T3SS while repress the expression of T4SS, T5SS, T6SS and flagella biogenesis genes. FleQ positively controls flagella biosynthesis and negatively regulates T6SS genes (26). The available nitrate from the extracellular space is imported into the cytoplasm by nitrogen-related transporter and reductase coding genes (*narA*, *nirBD*, *nark*, and *amtB*) which are regulated by the master regulator RpoN ($\sigma 54$). RpoN positively regulate the expression of T3SS and T6SS genes. Under phosphate limited condition the histidine kinase PhoR interacts with PstABC-PhoU complex and result in phosphorylation of PhoB which positively regulate the expression of T6SS genes (77, 78). At low concentration of zinc, the Zur regulator induces the expression of ZnuABC and activates T6SS genes (79). Breakdown of arginine by catabolic enzymes (SpeA and AguA) produce putrescine, exported out of the cell by transporters (PotABC) and helps in ROS scavenging (80).

In the present study, the expression of genes involved in T3SS and T3E were downregulated in the mutant strain, which was intriguing given the observation of hypervirulent phenotype with mutant *in planta* (51). We speculate one reason for decreased transcription level of T3SS genes could be the existence of a putative crosstalk between T6SS and T3SS as reported previously. Deletion of single T6SS resulted in decreased expression of T3SS genes in *Erwinia amylovora* (81), and *Pseudomonas syringae* pv. *actinidiae* (82). Recently, Li et al.

(83) reported similar results in XVM2 medium where the expression of *hrp* genes was decreased in *Xanthomonas campestris* pv. *campestris* *hrpA* deletion mutant. However, a contrasting expression pattern was observed when the pathogen was infected to the plant. Although, the minimal medium XVM2, which is suspected to mimic the plant apoplastic environment, was used as an alternative system for transcriptome expression studies under controlled conditions, it lacks specific plant signals that triggers the *hrp* regulon *in vitro*. The induction of T3SS genes is much stronger in the presence of plant-derived signals than in minimal medium (84, 85). In *P. syringae* DC3000 strain, exposure to tomato exudates activates the expression of *hrpK* suggesting that the phytopathogen senses the soluble low-molecular weight host signals in the exudate resulting in activation of T3SS genes (86). However, the mechanism by which the pathogen senses the plant signal and the regulatory mechanism that contributes to the induction of host factors and virulence factors in pathogens is unclear. This result and our findings imply that the mechanism of *hrp* genes activation and regulation under *in vitro* *hrp*-inducing conditions is divergent from that *in planta*. Regardless, the present study provided us with the understanding of the regulatory network of TssM without the complexity of spatial regulation observed *in planta*. Under such simplified conditions, inactivation of TssM leading to decreased T3SS gene expression suggest intertwined network with direct or indirect interaction of T6SS and T3SS.

One striking result observed in this RNA-Seq analysis was the significant upregulation of T6SS genes belonging to both clusters *i3** and *i3**** in the *tssM* mutant. The presence of multiple T6SS has been reported in several plant pathogens and each could be regulated differently and might carry out diverse functions as evident in other phytopathogens. In *X. citri*, T6SS-*i3** was induced in response to predation by soil-dwelling amoeba *D. discoideum* (11). The expression of T6SS-*i3** was regulated by a series of signaling cascade events modulated by a Serine/Threonine kinase, which was also upregulated in our study. This is important as the phyllosphere serves as a habitat for bacterial predators, induction of T6SS-*i3** helps to compete for niche during epiphytic colonization. In contrast, T6SS-*i3**** subgroup of *X. oryzae* pv. *oryzicola* was induced within the plant host revealed by transcriptome study but its function remains unclear (87). In *Pantoea ananatis* LMG 2665, a major pathogen of pineapple and onion, of the two T6SSs identified, T6SS-1 was found to be responsible for pathogenicity and competitiveness while T6SS-3 had no effect on virulence of the strain (10). Similarly, mutants in T6SS-1 and T6SS-2 genes of *Xanthomonas oryzae* pv. *oryzicola* resulted in no significant difference in virulence when compared to the wild type

but was required for bacterial competition (14). It is still elusive how various pathways integrate to activate the T6SS machinery (88). Overall, this supports the notion that different phytopathogen have specific regulatory networks to deploy their T6SS in response to different environments signals. Given our observation of increased expression of genes from both T6SS cluster may indicate a crosstalk among the two clusters.

Many genes associated with polysaccharide production and its regulation were highly upregulated in the mutant strain. High molecular weight extracellular polysaccharides and glycoproteins are key components of bacterial cell wall. In pathogenic bacteria, these molecules are crucial players in evading host immune responses, colonization, stress response and biofilm formation (89, 90). In *Xylella fastidiosa*, O-antigen delays host immune response in grape xylem (91). Similarly, several genes with glycosyltransferase were essential for fitness of *P. syringae* during leaf and apoplastic colonization (92). We reasoned that the increased expression of polysaccharide could be attributed to the increased expression of the global regulator *csrA* in the mutant strain. In xanthomonads, production of exopolysaccharide xanthan gum is positively regulated by CsrA (93), which contributes to proliferation of bacteria within intracellular apoplastic space resulting in development of disease symptoms (94, 95). The regulatory network of CsrA/RsmA is poorly understood in *Xanthomonas* spp. It is interesting that RsmA/CsrA negatively regulates genes associated with flagellum biogenesis, T4SS, T5SS and T6SS, which are essential for epiphytic colonization of pathogen, while positively regulating T3SS genes required for apoplastic colonization. This may imply that CsrA could possibly function as a molecular switch in *Xanthomonas* spp. shifting from epiphytic to endophytic lifestyle. Another major regulator, the alternative sigma factor RpoN that influences biofilm formation, virulence and growth displayed increased expression in the mutant strain (96–100). The role of RpoN in regulating T6SS is well established in animal pathogens (24, 101, 102) but in phytobacteria, very little is known, which opens an area for further investigation.

During colonization of host, the pathogen is constantly exposed to abiotic stresses. Responding these stresses plays a vital role in the fitness of the organism. In this transcriptome analysis, transcript abundance of several genes implicated with polyamine biosynthesis and transport were significantly upregulated in the mutant strain. In phytopathogens, synthesis and secretion of putrescine was reported in *R. solanacearum*, *P. syringae*, and *Dickeya zeae* (103–105), which play an important role in alleviating the oxidative stress by scavenging the

reactive oxygen species (106, 107). Further, two genes (*osmC* and *ompW*) associated with osmotic stress were significantly down regulated in the mutant strain which was in concordance with our previous result in which the *tssM* mutant displayed reduced growth when exposed to osmotic stress (51). These findings suggest the role of functional TssM in stress tolerance.

In summary, our data demonstrate that a highly intricate signaling pathway orchestrated by multiple regulators is involved in the regulation of T6SS genes. We observed that expression of major virulence factor regulators like c-di-GMP metabolic genes, sigma factors (RpoN, RpoE and FliA), inorganic ion homeostasis genes (Pho regulon and Zur), RNA binding protein CsrA were controlled by TssM either directly or indirectly, as shown in the schematic representation of the regulatory circuit mediated by *tssM* gene (Fig. 5). Based on our study and from previous works, we conclude that regulatory networks modulating virulence factors are highly complex in *Xanthomonas* and crosstalk among T6SS and other secretion systems may govern virulence, stress tolerance and metabolism. Our findings also highlight the importance of *in vitro* transcriptomics to understand basal regulatory networks that are otherwise highly context-dependent, be it host-derived cues or spatial expression.

Materials and Methods

Bacterial strains and growth conditions

Xanthomonas perforans wild type (WT) strain AL65 and its isogenic *tssM* mutant were routinely cultured on nutrient agar (NA) (Difco, USA) plates at 28°C. Liquid cultures were grown to the exponential phase at 28°C with continuous shaking at 150 rpm. For RNA-Seq experiment, *X. perforans* strains were grown in XVM2 medium (10 mM sucrose, 10 mM fructose, 0.01 mM casamino acids, 10 mM (NH₄)₂SO₄, 5 mM MgSO₄, 0.16 mM K₂HPO₄, 0.32 mM KH₂PO₄, 20 mM NaCl, 1 mM CaCl₂, 0.01 mM FeSO₄, pH 6.7) (108) with initial inoculum concentration of 10⁵ cfu/ml. Both wild type and the mutant strains were grown at 28°C with agitation. The bacterial cells were collected post growth in XVM2 medium at 8 h and 16 h post-inoculation.

RNA extraction

Total RNA was extracted using TRIzol™ (Thermo Fisher Scientific, Waltham, MA, USA) protocol (109). Four independent RNA extractions were obtained for each strain for each

time point. The concentration of RNA samples was determined using Qubit™ RNA HS Assay kit (Invitrogen). Ribosomal RNA was depleted using the MICROB Express Bacterial mRNA purification kit (Ambion, Foster City, CA, USA). The integrity and quality of the RNA was checked by an Agilent 2100 Bioanalyzer (Agilent Technologies, Santa Clara, CA, USA).

RNA-Seq data processing and differential expression analysis

The RNA sequencing was performed at MiGS using Illumina NextSeq 2000 to generate 12M, 2x51bp reads/sample. The sequenced data was processed to generate FASTQ files. Two pipelines were used to calculate differential gene expression (DGE) to account for variation observed depending on models used for transcript estimation (110). The first pipeline used was epinatel (https://github.com/epinatel/Bacterial_RNAseq/). The second pipeline used for DGE analysis was brnaseq (<https://github.com/barricklab/brnaseq/blob/master/README.md>). Both pipelines used steps involving read quality filtering using trimmomatic (111), mapping to the reference genome of *X. perforans* AL65 (RefSeq: NZ_SMVI00000000.1, Assembly: GCF_007714115.1) using bowtie2 (112), DGE using DESeq2 (113), except for read counting step with the first pipeline using custom scripts and second using HTSeq (114). Significantly up-or-down regulated genes were identified as those in which the absolute value of log (in base 2) of fold-change was greater than 0.58 and the adjusted false discovery rate (FDR) was ≤ 0.05 . The output of two pipelines was compared. The data obtained from epinatel is represented in tables and figures (after confirming no difference in top differentially expressed genes (DEGs) obtained in both methods). Gene ontology analysis was performed using EggNOG v5.0 (115).

Motility

The ability of wild type and mutant strains to swim and twitch was screened on NA and XVM2 agar. Single colonies of *X. perforans* AL65 wild type and AL65 Δ tssM mutant were inoculated into nutrient broth and XVM2 medium and grown for 24 h at 28°C. Bacterial growth was then normalized using respective mediums to 10⁸ CFU/mL (OD_{600nm} = 0.3) using spectrophotometer (Thermo Fisher Scientific GENESYS 10S UV-Vis). For each strain, 2 μ L of the suspension was placed onto the centre of the plates and incubated at room temperature for 48 h without agitation. The swimming halos were measured to identify the difference in the motility between wild type and the mutant strain. For twitching motility, the fringe width

was visualised using microscope (Revolve Microscope, Echo Laboratories, San Diego, CA). The experiment was performed in triplicates.

Biofilm Assay

The wild type and the mutant strains were grown overnight at 28°C in XVM2 medium and normalized to an OD_{600nm} 0.3 to access the in-vitro biofilm formation on abiotic surface. The glass tubes were incubated for 24 and 48 h at 28 °C without agitation. The planktonic cells were carefully aspirated, and the optical density was measured at 600nm to access the growth of both strains using spectrophotometer (Thermo Fisher Scientific GENESYS 10S UV-Vis). The loosely adhered cells were removed by washing the tubes carefully with MQ three times. The sessile cells were stained with 3 ml of 0.1% crystal violet solution. The tubes were incubated for 30 minutes without shaking at room temperature. After incubation, the crystal violet stain was removed carefully, and excess stain was washed off with MQ. For quantification of biofilm, the stain was solubilized using a solution of 40% methanol and 10% glacial acetic acid. Absorbance was measured at O.D._{570 nm}. Eight biological replicates of each strain were performed.

Exopolysaccharide (EPS) Assay

The ability of wild type and the mutant strain to produce EPS was estimated as previously described (116). Briefly, overnight culture of the wild type and *tssM* mutant strains was normalized to an OD_{600nm} of 0.3 in XVM2 media and grown at 28°C with agitation for 48 h. The bacterial cells were pelleted at 6000g for 10 mins. The supernatant was carefully transferred to a fresh tube. To the supernatant two volumes of ice-cold 96% ethanol (v/v) was added to precipitate EPS, the tubes were incubated at 4°C overnight. After precipitation, the samples were centrifuged at 6000g for 30 min at 4°C. The supernatant was carefully discarded, and the pellet was air dried at room temperature for overnight. Dried pellets were weighed to quantify EPS produced by wild type and the mutant strain. Three biological replicates of each strain were performed.

Data Availability

The RNA-Seq reads used in this study are deposited in the National Center for Biotechnology Information sequence read archive database (Bioproject: PRJNA954348). The scripts, input files, intermediate files and output files used for interpretation of the analyses conducted in this work are made available on https://github.com/Potnislav/T6SS_RNASeq_manuscript

as well as [DOI: 10.5281/zenodo.8144145](https://doi.org/10.5281/zenodo.8144145) (117).

Acknowledgement

We thank David Young and the staff at Alabama Supercomputer authority for providing the computational resources necessary for this work. Additionally, we thank 
This work was supported by the Alabama Ag Experiment Station at Auburn University and the National Science Foundation, Division of Integrative Organismal Systems (IOS-1942956) project to Neha Potnis.

References cited

1. Stanley SA, Raghavan S, Hwang WW, Cox JS. 2003. Acute infection and macrophage subversion by *Mycobacterium tuberculosis* require a specialized secretion system. *Proceedings of the National Academy of Sciences* 100:13001–13006.
2. Whitney JC, Howell PL. 2013. Synthase-dependent exopolysaccharide secretion in Gram-negative bacteria. *Trends in Microbiology* 21:63–72.
3. Kneuper H, Cao ZP, Twomey KB, Zoltner M, Jäger F, Cargill JS, Chalmers J, van der Kooi-Pol MM, van Dijl JM, Ryan RP, Hunter WN, Palmer T. 2014. Heterogeneity in *Ess* transcriptional organization and variable contribution of the *Ess*/Type VII protein secretion system to virulence across closely related *Staphylococcus aureus* strains. *Molecular Microbiology* 93:928–943.
4. Green ER, Meccas J. 2016. Bacterial Secretion Systems: An Overview. *Microbiol Spectr* 4.
5. Boyer F, Fichant G, Berthod J, Vandenbrouck Y, Attree I. 2009. Dissecting the bacterial type VI secretion system by a genome wide in silico analysis: what can be learned from available microbial genomic resources? *BMC Genomics* 10:104.
6. Habbadi K, Benkirane R, Benbouazza A, Bouaichi A, Maafa I, Chapulliot D, Achbani E. 2017. Biological Control of Grapevine Crown Gall Caused by *Allorhizobium vitis* using Bacterial Antagonists.
7. Wu C-F, Smith DA, Lai E-M, Chang JH. 2018. The Agrobacterium Type VI Secretion System: A Contractile Nanomachine for Interbacterial Competition, p. 215–231. *In* Gelvin, SB (ed.), *Agrobacterium Biology: From Basic Science to Biotechnology*. Springer International Publishing, Cham.
8. Haapalainen M, Mosorin H, Dorati F, Wu R-F, Roine E, Taira S, Nissinen R, Mattinen L, Jackson R, Pirhonen M, Lin N-C. 2012. Hcp2, a Secreted Protein of the Phytopathogen *Pseudomonas syringae* pv. *Tomato* DC3000, Is Required for Fitness for Competition against Bacteria and Yeasts. *Journal of Bacteriology* 194:4810–4822.
9. Ma L-S, Hachani A, Lin J-S, Filloux A, Lai E-M. 2014. Agrobacterium tumefaciens Deploys a Superfamily of Type VI Secretion DNase Effectors as Weapons for Interbacterial Competition In Planta. *Cell Host Microbe* 16:94–104.

10. Shyntum DY, Theron J, Venter SN, Moleleki LN, Toth IK, Coutinho TA. 2015. *Pantoea ananatis* Utilizes a Type VI Secretion System for Pathogenesis and Bacterial Competition. *MPMI* 28:420–431.
11. Bayer-Santos E, Lima L dos P, Ceseti L de M, Ratagami CY, de Santana ES, da Silva AM, Farah CS, Alvarez-Martinez CE. 2018. *Xanthomonas citri* T6SS mediates resistance to *Dictyostelium* predation and is regulated by an ECF σ factor and cognate Ser/Thr kinase. *Environmental Microbiology* 20:1562–1575.
12. Montenegro Benavides NA, Alvarez B. A, Arrieta-Ortiz ML, Rodriguez-R LM, Botero D, Tabima JF, Castiblanco L, Trujillo C, Restrepo S, Bernal A. 2021. The type VI secretion system of *Xanthomonas phaseoli* pv. *manihotis* is involved in virulence and in vitro motility. *BMC Microbiology* 21:14.
13. Bayer-Santos E, Ceseti L de M, Farah CS, Alvarez-Martinez CE. 2019. Distribution, Function and Regulation of Type 6 Secretion Systems of Xanthomonadales. *Frontiers in Microbiology* 10.
14. Zhu P-C, Li Y-M, Yang X, Zou H-F, Zhu X-L, Niu X-N, Xu L-H, Jiang W, Huang S, Tang J-L, He Y-Q. 2020. Type VI secretion system is not required for virulence on rice but for inter-bacterial competition in *Xanthomonas oryzae* pv. *oryzicola*. *Research in Microbiology* 171:64–73.
15. Ho BT, Dong TG, Mekalanos JJ. 2014. A view to a kill: the bacterial type VI secretion system. *Cell Host Microbe* 15:9–21.
16. Ma AT, Mekalanos JJ. 2010. In vivo actin cross-linking induced by *Vibrio cholerae* type VI secretion system is associated with intestinal inflammation. *Proceedings of the National Academy of Sciences* 107:4365–4370.
17. Bondage DD, Lin J-S, Ma L-S, Kuo C-H, Lai E-M. 2016. VgrG C terminus confers the type VI effector transport specificity and is required for binding with PAAR and adaptor-effector complex. *Proc Natl Acad Sci U S A* 113:E3931-3940.
18. Cianfanelli FR, Monlezun L, Coulthurst SJ. 2016. Aim, Load, Fire: The Type VI Secretion System, a Bacterial Nanoweapon. *Trends Microbiol* 24:51–62.
19. Flaugnatti N, Le TTH, Canaan S, Aschtgen M-S, Nguyen VS, Blangy S, Kellenberger C, Roussel A, Cambillau C, Cascales E, Journet L. 2016. A phospholipase A1 antibacterial Type VI secretion effector interacts directly with the C-terminal domain of the VgrG spike protein for delivery. *Mol Microbiol* 99:1099–1118.
20. Pukatzki S, Ma AT, Revel AT, Sturtevant D, Mekalanos JJ. 2007. Type VI secretion system translocates a phage tail spike-like protein into target cells where it cross-links actin. *Proceedings of the National Academy of Sciences* 104:15508–15513.
21. Durand E, Zoued A, Spinelli S, Watson PJH, Aschtgen M-S, Journet L, Cambillau C, Cascales E. 2012. Structural Characterization and Oligomerization of the TssL Protein, a Component Shared by Bacterial Type VI and Type IVb Secretion Systems *. *Journal of Biological Chemistry* 287:14157–14168.
22. Whitney JC, Beck CM, Goo YA, Russell AB, Harding BN, De Leon JA, Cunningham DA, Tran BQ, Low DA, Goodlett DR, Hayes CS, Mougous JD. 2014. Genetically distinct pathways guide effector export through the type VI secretion system. *Mol Microbiol* 92:529–542.

23. Quentin D, Ahmad S, Shanthamoorthy P, Mougous JD, Whitney JC, Raunser S. 2018. Mechanism of loading and translocation of type VI secretion system effector Tse6. *Nat Microbiol* 3:1142–1152.
24. Dong TG, Mekalanos JJ. 2012. Characterization of the RpoN regulon reveals differential regulation of T6SS and new flagellar operons in *Vibrio cholerae* O37 strain V52. *Nucleic Acids Res* 40:7766–7775.
25. Allsopp LP, Collins ACZ, Hawkins E, Wood TE, Filloux A. 2022. RpoN/Sfa2-dependent activation of the *Pseudomonas aeruginosa* H2-T6SS and its cognate arsenal of antibacterial toxins. *Nucleic Acids Research* 50:227–243.
26. Wang Y, Li Y, Wang J, Wang X. 2018. FleQ regulates both the type VI secretion system and flagella in *Pseudomonas putida*. *Biotechnology and Applied Biochemistry* 65:419–427.
27. Castang S, McManus HR, Turner KH, Dove SL. 2008. H-NS family members function coordinately in an opportunistic pathogen. *Proceedings of the National Academy of Sciences* 105:18947–18952.
28. Wang J, Brodmann M, Basler M. 2019. Assembly and Subcellular Localization of Bacterial Type VI Secretion Systems. *Annual Review of Microbiology* 73:621–638.
29. Brunet YR, Khodr A, Logger L, Aussel L, Mignot T, Rimsky S, Cascales E. 2015. H-NS Silencing of the Salmonella Pathogenicity Island 6-Encoded Type VI Secretion System Limits *Salmonella enterica* Serovar *Typhimurium* Interbacterial Killing. *Infection and Immunity* 83:2738–2750.
30. Silverman JM, Brunet YR, Cascales E, Mougous JD. 2012. Structure and Regulation of the Type VI Secretion System. *Annual Review of Microbiology* 66:453–472.
31. Zheng J, Shin OS, Cameron DE, Mekalanos JJ. 2010. Quorum sensing and a global regulator TsrA control expression of type VI secretion and virulence in *Vibrio cholerae*. *Proceedings of the National Academy of Sciences* 107:21128–21133.
32. Liu X, Pan J, Gao H, Han Y, Zhang A, Huang Y, Liu P, Kan B, Liang W. 2021. CqsA/LuxS-HapR Quorum sensing circuit modulates type VI secretion system VftT6SS2 in *Vibrio fluvialis*. *Emerg Microbes Infect* 10:589–601.
33. Sana TG, Hachani A, Bucior I, Soscia C, Garvis S, Termine E, Engel J, Filloux A, Bleves S. 2012. The second type VI secretion system of *Pseudomonas aeruginosa* strain PAO1 is regulated by quorum sensing and Fur and modulates internalization in epithelial cells. *J Biol Chem* 287:27095–27105.
34. Majerczyk C, Schneider E, Greenberg EP. 2016. Quorum sensing control of Type VI secretion factors restricts the proliferation of quorum-sensing mutants. *eLife* 5:e14712.
35. Khajanchi BK, Sha J, Kozlova EV, Erova TE, Suarez G, Sierra JC, Popov VL, Horneman AJ, Chopra AK. 2009. N-Acylhomoserine lactones involved in quorum sensing control the type VI secretion system, biofilm formation, protease production, and in vivo virulence in a clinical isolate of *Aeromonas hydrophila*. *Microbiology (Reading)* 155:3518–3531.
36. Brencic A, Lory S. 2009. Determination of the regulon and identification of novel mRNA targets of *Pseudomonas aeruginosa* RsmA. *Molecular Microbiology* 72:612–632.

37. Basler M, Ho BT, Mekalanos JJ. 2013. Tit-for-Tat: Type VI Secretion System Counterattack during Bacterial Cell-Cell Interactions. *Cell* 152:884–894.
38. Mougous JD, Gifford CA, Ramsdell TL, Mekalanos JJ. 2007. Threonine phosphorylation post-translationally regulates protein secretion in *Pseudomonas aeruginosa*. 7. *Nat Cell Biol* 9:797–803.
39. Hsu F, Schwarz S, Mougous JD. 2009. TagR promotes PpkA-catalysed type VI secretion activation in *Pseudomonas aeruginosa*. *Molecular Microbiology* 72:1111–1125.
40. Bernal P, Allsopp LP, Filloux A, Llamas MA. 2017. The *Pseudomonas putida* T6SS is a plant warden against phytopathogens. 4. *ISME J* 11:972–987.
41. Ishikawa T, Sabharwal D, Bröms J, Milton DL, Sjöstedt A, Uhlin BE, Wai SN. 2012. Pathoadaptive conditional regulation of the type VI secretion system in *Vibrio cholerae* O1 strains. *Infect Immun* 80:575–584.
42. Townsley L, Sison Mangus MP, Mehic S, Yildiz FH. 2016. Response of *Vibrio cholerae* to Low-Temperature Shifts: CspV Regulation of Type VI Secretion, Biofilm Formation, and Association with Zooplankton. *Applied and Environmental Microbiology* 82:4441–4452.
43. Si M, Zhao C, Burkinshaw B, Zhang B, Wei D, Wang Y, Dong TG, Shen X. 2017. Manganese scavenging and oxidative stress response mediated by type VI secretion system in *Burkholderia thailandensis*. *Proceedings of the National Academy of Sciences* 114:E2233–E2242.
44. Si M, Wang Y, Zhang B, Zhao C, Kang Y, Bai H, Wei D, Zhu L, Zhang L, Dong TG, Shen X. 2017. The Type VI Secretion System Engages a Redox-Regulated Dual-Functional Heme Transporter for Zinc Acquisition. *Cell Reports* 20:949–959.
45. Li C, Zhu L, Wang D, Wei Z, Hao X, Wang Z, Li T, Zhang L, Lu Z, Long M, Wang Y, Wei G, Shen X. 2022. T6SS secretes an LPS-binding effector to recruit OMVs for exploitative competition and horizontal gene transfer. 2. *ISME J* 16:500–510.
46. Lin J, Zhang W, Cheng J, Yang X, Zhu K, Wang Y, Wei G, Qian P-Y, Luo Z-Q, Shen X. 2017. A *Pseudomonas* T6SS effector recruits PQS-containing outer membrane vesicles for iron acquisition. 1. *Nat Commun* 8:14888.
47. Yang X, Liu H, Zhang Y, Shen X. 2021. Roles of Type VI Secretion System in Transport of Metal Ions. *Frontiers in Microbiology* 12.
48. Miyata ST, Unterwieser D, Rudko SP, Pukatzki S. 2013. Dual Expression Profile of Type VI Secretion System Immunity Genes Protects Pandemic *Vibrio cholerae*. *PLOS Pathogens* 9:e1003752.
49. Yu K-W, Xue P, Fu Y, Yang L. 2021. T6SS Mediated Stress Responses for Bacterial Environmental Survival and Host Adaptation. 2. *International Journal of Molecular Sciences* 22:478.
50. Liyanapathirana P, Wagner N, Avram O, Pupko T, Potnis N. 2022. Phylogenetic Distribution and Evolution of Type VI Secretion System in the Genus *Xanthomonas*. *Front Microbiol* 13:840308.
51. Liyanapathirana P, Jones JB, Potnis N. 2022. Mutation of a Single Core Gene, *tssM*, of Type VI Secretion System of *Xanthomonas perforans* Influences Virulence,

- Epiphytic Survival, and Transmission During Pathogenesis on Tomato. *Phytopathology*® 112:752–764.
52. Zhang L, Xu J, Xu J, Zhang H, He L, Feng J. 2014. TssB is essential for virulence and required for Type VI secretion system in *Ralstonia solanacearum*. *Microbial Pathogenesis* 74:1–7.
 53. Tian Y, Zhao Y, Wu X, Liu F, Hu B, Walcott RR. 2015. The type VI protein secretion system contributes to biofilm formation and seed-to-seedling transmission of *Acidovorax citrulli* on melon. *Molecular Plant Pathology* 16:38–47.
 54. Records AR, Gross DC. 2010. Sensor Kinases RetS and LadS Regulate *Pseudomonas syringae* Type VI Secretion and Virulence Factors. *J Bacteriol* 192:3584–3596.
 55. Wu H-Y, Chung P-C, Shih H-W, Wen S-R, Lai E-M. 2008. Secretome Analysis Uncovers an Hcp-Family Protein Secreted via a Type VI Secretion System in *Agrobacterium tumefaciens*. *Journal of Bacteriology* 190:2841–2850.
 56. Abendroth U, Adlung N, Otto A, Grüneisen B, Becher D, Bonas U. 2017. Identification of new protein-coding genes with a potential role in the virulence of the plant pathogen *Xanthomonas euvesicatoria*. *BMC Genomics* 18:625.
 57. Ceseti LM, de Santana ES, Ratagami CY, Barreiros Y, Lima LDP, Dunger G, Farah CS, Alvarez-Martinez CE. 2019. The *Xanthomonas citri* pv. *citri* Type VI Secretion System is Induced During Epiphytic Colonization of Citrus. *Curr Microbiol* 76:1105–1111.
 58. Freeman BC, Chen C, Yu X, Nielsen L, Peterson K, Beattie GA. 2013. Physiological and Transcriptional Responses to Osmotic Stress of Two *Pseudomonas syringae* Strains That Differ in Epiphytic Fitness and Osmotolerance. *Journal of Bacteriology* 195:4742–4752.
 59. Coulthurst S. 2019. The Type VI secretion system: a versatile bacterial weapon. *Microbiology (Reading)* 165:503–515.
 60. Kim S, Cho Y-J, Song E-S, Lee SH, Kim J-G, Kang L-W. 2016. Time-resolved pathogenic gene expression analysis of the plant pathogen *Xanthomonas oryzae* pv. *oryzae*. *BMC Genomics* 17:345.
 61. Tsuge S, Furutani A, Fukunaka R, Oku T, Tsuno K, Ochiai H, Inoue Y, Kaku H, Kubo Y. 2002. Expression of *Xanthomonas oryzae* pv. *oryzae* hrp Genes in XOM2, a Novel Synthetic Medium. *J Gen Plant Pathol* 68:363–371.
 62. Seo Y-S, Sriariyanun M, Wang L, Pfeiff J, Phetsom J, Lin Y, Jung K-H, Chou HH, Bogdanove A, Ronald P. 2008. A two-genome microarray for the rice pathogens *Xanthomonas oryzae* pv. *oryzae* and *X. oryzae* pv. *oryzicola* and its use in the discovery of a difference in their regulation of hrp genes. *BMC Microbiol* 8:99.
 63. Guo Y, Figueiredo F, Jones J, Wang N. 2011. HrpG and HrpX Play Global Roles in Coordinating Different Virulence Traits of *Xanthomonas axonopodis* pv. *citri*. *MPMI* 24:649–661.
 64. Xie Y, Shao X, Deng X. 2019. Regulation of type III secretion system in *Pseudomonas syringae*. *Environmental Microbiology* 21:4465–4477.
 65. Chen Y, Wong J, Sun GW, Liu Y, Tan G-YG, Gan Y-H. 2011. Regulation of type VI secretion system during *Burkholderia pseudomallei* infection. *Infect Immun* 79:3064–3073.

66. Burtnick MN, Brett PJ, Harding SV, Ngugi SA, Ribot WJ, Chantratita N, Scorpio A, Milne TS, Dean RE, Fritz DL, Peacock SJ, Prior JL, Atkins TP, DeShazer D. 2011. The Cluster 1 Type VI Secretion System Is a Major Virulence Determinant in *Burkholderia pseudomallei*. *Infect Immun* 79:1512–1525.
67. Viberg LT, Sarovich DS, Kidd TJ, Geake JB, Bell SC, Currie BJ, Price EP. 2017. Within-Host Evolution of *Burkholderia pseudomallei* during Chronic Infection of Seven Australasian Cystic Fibrosis Patients. *mBio* 8:e00356-17.
68. Gilston BA, Wang S, Marcus MD, Canalizo-Hernández MA, Swindell EP, Xue Y, Mondragón A, O'Halloran TV. 2014. Structural and Mechanistic Basis of Zinc Regulation Across the *E. coli* Zur Regulon. *PLOS Biology* 12:e1001987.
69. Wang J-Y, Zhou L, Chen B, Sun S, Zhang W, Li M, Tang H, Jiang B-L, Tang J-L, He Y-W. 2015. A functional 4-hydroxybenzoate degradation pathway in the phytopathogen *Xanthomonas campestris* is required for full pathogenicity. 1. *Sci Rep* 5:18456.
70. Lee SK, Stack A, Katzowitsch E, Aizawa SI, Suerbaum S, Josenhans C. 2003. *Helicobacter pylori* flagellins have very low intrinsic activity to stimulate human gastric epithelial cells via TLR5. *Microbes and Infection* 5:1345–1356.
71. Gómez-Gómez L, Boller T. 2000. FLS2: An LRR Receptor-like Kinase Involved in the Perception of the Bacterial Elicitor Flagellin in *Arabidopsis*. *Molecular Cell* 5:1003–1011.
72. Haiko J, Westerlund-Wikström B. 2013. The Role of the Bacterial Flagellum in Adhesion and Virulence. *Biology (Basel)* 2:1242–1267.
73. Yang T-C, Leu Y-W, Chang-Chien H-C, Hu R-M. 2009. Flagellar Biogenesis of *Xanthomonas campestris* Requires the Alternative Sigma Factors RpoN2 and FliA and Is Temporally Regulated by FlhA, FlhB, and FlgM. *J Bacteriol* 191:2266–2275.
74. Hsiao Y-M, Liu Y-F, Fang M-C, Song W-L. 2011. XCC2731, a GGDEF domain protein in *Xanthomonas campestris*, is involved in bacterial attachment and is positively regulated by Clp. *Microbiol Res* 166:548–565.
75. Su J, Zou X, Huang L, Bai T, Liu S, Yuan M, Chou S-H, He Y-W, Wang H, He J. 2016. DgcA, a diguanylate cyclase from *Xanthomonas oryzae* pv. *oryzae* regulates bacterial pathogenicity on rice. 1. *Sci Rep* 6:25978.
76. Büttner D, Bonas U. 2010. Regulation and secretion of *Xanthomonas* virulence factors. *FEMS Microbiol Rev* 34:107–133.
77. Chakraborty S, Sivaraman J, Leung KY, Mok Y-K. 2011. Two-component PhoB-PhoR Regulatory System and Ferric Uptake Regulator Sense Phosphate and Iron to Control Virulence Genes in Type III and VI Secretion Systems of *Edwardsiella tarda**. *Journal of Biological Chemistry* 286:39417–39430.
78. Chekabab SM, Harel J, Dozois CM. 2014. Interplay between genetic regulation of phosphate homeostasis and bacterial virulence. *Virulence* 5:786–793.
79. Cai R, Gao F, Pan J, Hao X, Yu Z, Qu Y, Li J, Wang D, Wang Y, Shen X, Liu X, Yang Y. 2021. The transcriptional regulator Zur regulates the expression of ZnuABC and T6SS4 in response to stresses in *Yersinia pseudotuberculosis*. *Microbiological Research* 249:126787.

80. Solmi L, Rosli HG, Pombo MA, Stalder S, Rossi FR, Romero FM, Ruiz OA, Gárriz A. 2022. Inferring the Significance of the Polyamine Metabolism in the Phytopathogenic Bacteria *Pseudomonas syringae*: A Meta-Analysis Approach. *Frontiers in Microbiology* 13.
81. Kamber T, Pothier JF, Pelludat C, Rezzonico F, Duffy B, Smits THM. 2017. Role of the type VI secretion systems during disease interactions of *Erwinia amylovora* with its plant host. *BMC Genomics* 18:628.
82. Wang N, Han N, Tian R, Chen J, Gao X, Wu Z, Liu Y, Huang L. 2021. Role of the Type VI Secretion System in the Pathogenicity of *Pseudomonas syringae* pv. *actinidiae*, the Causative Agent of Kiwifruit Bacterial Canker. *Frontiers in Microbiology* 12.
83. Li R, Ren P, Zhang D, Cui P, Zhu G, Xian X, Tang J, Lu G. 2022. HpaP divergently regulates the expression of *hrp* genes in *Xanthomonas oryzae* pathovars *oryzae* and *oryzicola*. *Mol Plant Pathol* 24:44–58.
84. Rahme LG, Mindrinos MN, Panopoulos NJ. 1992. Plant and environmental sensory signals control the expression of *hrp* genes in *Pseudomonas syringae* pv. *phaseolicola*. *J Bacteriol* 174:3499–3507.
85. Yahr TL, Wolfgang MC. 2006. Transcriptional regulation of the *Pseudomonas aeruginosa* type III secretion system. *Mol Microbiol* 62:631–640.
86. Haapalainen M, van Gestel K, Pirhonen M, Taira S. 2009. Soluble Plant Cell Signals Induce the Expression of the Type III Secretion System of *Pseudomonas syringae* and Upregulate the Production of Pilus Protein HrpA. *MPMI* 22:282–290.
87. Liao Z-X, Ni Z, Wei X-L, Chen L, Li J-Y, Yu Y-H, Jiang W, Jiang B-L, He Y-Q, Huang S. 2019. Dual RNA-seq of *Xanthomonas oryzae* pv. *oryzicola* infecting rice reveals novel insights into bacterial-plant interaction. *PLOS ONE* 14:e0215039.
88. Bernard CS, Brunet YR, Gueguen E, Cascales E. 2010. Nooks and crannies in type VI secretion regulation. *J Bacteriol* 192:3850–3860.
89. Yakovlieva L, Walvoort MTC. 2020. Processivity in Bacterial Glycosyltransferases. *ACS Chem Biol* 15:3–16.
90. Erbs G, Newman M-A. 2012. The role of lipopolysaccharide and peptidoglycan, two glycosylated bacterial microbe-associated molecular patterns (MAMPs), in plant innate immunity. *Mol Plant Pathol* 13:95–104.
91. Rapicavoli JN, Blanco-Ulate B, Muszyński A, Figueroa-Balderas R, Morales-Cruz A, Azadi P, Dobruchowska JM, Castro C, Cantu D, Roper MC. 2018. Lipopolysaccharide O-antigen delays plant innate immune recognition of *Xylella fastidiosa*. 1. *Nat Commun* 9:390.
92. Helmann TC, Deutschbauer AM, Lindow SE. 2019. Genome-wide identification of *Pseudomonas syringae* genes required for fitness during colonization of the leaf surface and apoplast. *Proc Natl Acad Sci U S A* 116:18900–18910.
93. Zhang X, Li N, Liu X, Wang J, Zhang Y, Liu D, Wang Y, Cao H, Zhao B, Yang W. 2021. Tomato protein Rx4 mediates the hypersensitive response to *Xanthomonas euvesicatoria* pv. *perforans* race T3. *The Plant Journal* 105:1630–1644.

94. Li J, Wang N. 2011. The wxacO gene of *Xanthomonas citri* ssp. *citri* encodes a protein with a role in lipopolysaccharide biosynthesis, biofilm formation, stress tolerance and virulence. *Mol Plant Pathol* 12:381–396.
95. Rigano LA, Siciliano F, Enrique R, Sendín L, Filippone P, Torres PS, Qüesta J, Dow JM, Castagnaro AP, Vojnov AA, Marano MR. 2007. Biofilm Formation, Epiphytic Fitness, and Canker Development in *Xanthomonas axonopodis* pv. *citri*. *MPMI* 20:1222–1230.
96. Hao B, Mo Z-L, Xiao P, Pan H-J, Lan X, Li G-Y. 2013. Role of alternative sigma factor 54 (RpoN) from *Vibrio anguillarum* M3 in protease secretion, exopolysaccharide production, biofilm formation, and virulence. *Appl Microbiol Biotechnol* 97:2575–2585.
97. Hayrapetyan H, Tempelaars M, Nierop Groot M, Abee T. 2015. *Bacillus cereus* ATCC 14579 RpoN (Sigma 54) Is a Pleiotropic Regulator of Growth, Carbohydrate Metabolism, Motility, Biofilm Formation and Toxin Production. *PLoS One* 10:e0134872.
98. Ray SK, Kumar R, Peeters N, Boucher C, Genin S. 2015. rpoN1, but not rpoN2, is required for twitching motility, natural competence, growth on nitrate, and virulence of *Ralstonia solanacearum*. *Front Microbiol* 6:229.
99. Li K, Wu G, Liao Y, Zeng Q, Wang H, Liu F. 2020. RpoN1 and RpoN2 play different regulatory roles in virulence traits, flagellar biosynthesis, and basal metabolism in *Xanthomonas campestris*. *Mol Plant Pathol* 21:907–922.
100. Hendrickson EL, Guevera P, Ausubel FM. 2000. The Alternative Sigma Factor RpoN Is Required for hrp Activity in *Pseudomonas syringae* pv. *Maculicola* and Acts at the Level of hrpL Transcription. *J Bacteriol* 182:3508–3516.
101. Shao X, Zhang X, Zhang Y, Zhu M, Yang P, Yuan J, Xie Y, Zhou T, Wang W, Chen S, Liang H, Deng X. 2018. RpoN-Dependent Direct Regulation of Quorum Sensing and the Type VI Secretion System in *Pseudomonas aeruginosa* PAO1. *Journal of Bacteriology* 200:e00205-18.
102. Storey D, McNally A, Åstrand M, Santos J sa-PG, Rodriguez-Escudero I, Elmore B, Palacios L, Marshall H, Hopley L, Molina M, Cid VJ, Salminen TA, Bengoechea JA. 2020. *Klebsiella pneumoniae* type VI secretion system-mediated microbial competition is PhoPQ controlled and reactive oxygen species dependent. *PLOS Pathogens* 16:e1007969.
103. Lowe-Power TM, Hendrich CG, von Roepenack-Lahaye E, Li B, Wu D, Mitra R, Dalsing BL, Ricca P, Naidoo J, Cook D, Jancewicz A, Masson P, Thomma B, Lahaye T, Michael AJ, Allen C. 2018. Metabolomics of tomato xylem sap during bacterial wilt reveals *Ralstonia solanacearum* produces abundant putrescine, a metabolite that accelerates wilt disease. *Environmental Microbiology* 20:1330–1349.
104. Vilas JM, Romero FM, Rossi FR, Marina M, Maiale SJ, Calzadilla PI, Pieckenstain FL, Ruiz OA, Gárriz A. 2018. Modulation of plant and bacterial polyamine metabolism during the compatible interaction between tomato and *Pseudomonas syringae*. *Journal of Plant Physiology* 231:281–290.
105. Shi Z, Wang Q, Li Y, Liang Z, Xu L, Zhou J, Cui Z, Zhang L-H. 2019. Putrescine Is an Intraspecies and Interkingdom Cell-Cell Communication Signal Modulating the Virulence of *Dickeya zaeae*. *Frontiers in Microbiology* 10.

106. Nayyar H, Chander S. 2004. Protective Effects of Polyamines against Oxidative Stress Induced by Water and Cold Stress in Chickpea. *Journal of Agronomy and Crop Science* 190:355–365.
107. Tang W, Newton RJ. 2005. Polyamines reduce salt-induced oxidative damage by increasing the activities of antioxidant enzymes and decreasing lipid peroxidation in Virginia pine. *Plant Growth Regul* 46:31–43.
108. Wengelnik K, Van den Ackerveken G, Bonas U. 1996. HrpG, a key hrp regulatory protein of *Xanthomonas campestris* pv. *vesicatoria* is homologous to two-component response regulators. *Mol Plant Microbe Interact* 9:704–712.
109. Rio DC, Ares M, Hannon GJ, Nilsen TW. 2010. Purification of RNA using TRIzol (TRI reagent). *Cold Spring Harb Protoc* 2010:pdb.prot5439.
110. Trapnell C, Hendrickson DG, Sauvageau M, Goff L, Rinn JL, Pachter L. 2013. Differential analysis of gene regulation at transcript resolution with RNA-seq. 1. *Nat Biotechnol* 31:46–53.
111. Bolger AM, Lohse M, Usadel B. 2014. Trimmomatic: a flexible trimmer for Illumina sequence data. *Bioinformatics* 30:2114–2120.
112. Langmead B, Salzberg SL. 2012. Fast gapped-read alignment with Bowtie 2. *Nat Methods* 9:357–359.
113. Love MI, Huber W, Anders S. 2014. Moderated estimation of fold change and dispersion for RNA-seq data with DESeq2. *Genome Biology* 15:550.
114. Anders S, Pyl PT, Huber W. 2015. HTSeq—a Python framework to work with high-throughput sequencing data. *Bioinformatics* 31:166–169.
115. Huerta-Cepas J, Szklarczyk D, Heller D, Hernández-Plaza A, Forslund SK, Cook H, Mende DR, Letunic I, Rattei T, Jensen LJ, von Mering C, Bork P. 2019. eggNOG 5.0: a hierarchical, functionally and phylogenetically annotated orthology resource based on 5090 organisms and 2502 viruses. *Nucleic Acids Research* 47:D309–D314.
116. Dabour N, Kheadr EE, Fliss I, LaPointe G. 2005. Impact of ropy and capsular exopolysaccharide-producing strains of *Lactococcus lactis* subsp. *cremoris* on reduced-fat Cheddar cheese production and whey composition. *International Dairy Journal* 15:459–471.
117. [dataset] Ramamoorthy S, Pena M, Ghosh P, Liao Y-Y, Paret M, Jones JB, Potnis N. 2023. Dataset for " Transcriptome profiling of type VI secretion system core gene tssM mutant of *Xanthomonas perforans* highlights regulators controlling diverse functions ranging from virulence to metabolism" [T6SS_RNASeq_analysis](https://doi.org/10.5281/zenodo.8144145), Zenodo, DOI: [10.5281/zenodo.8144145](https://doi.org/10.5281/zenodo.8144145)

Figure Legends

Figure 1. *In vitro* transcriptome profiling revealed nearly 14% and 22% of genes were differentially expressed in the *tssM* mutant post 8 and 16 h of growth in XVM2 medium.

(A) Volcano plot illustrating the upregulated and downregulated genes in *tssM* mutant at 8 h (left) and 16 h (right). Each dot indicates an individual coding sequence with detectable expression in the mutant strain in comparison to the wild type. The X-axis represents the log₂ fold change between mutant and wild type, and the Y-axis represents the $-\log_{10}$ transformed adjusted *p*-value denoting the significance of differential expression. (B) Venn diagram of genes with significant difference in transcript level between mutant and wild type. The genes showing overlap between the two time points are represented in percentage. (C) Functional categorisation of differential expressed genes in the *tssM* mutant based on the clusters of orthologous groups (COGs).

Figure 2. Deletion of *tssM* resulted in downregulation of flagella biosynthesis genes. (A)

Heat map showing expression pattern of genes involved in flagella production and chemotaxis. (B) Swimming motility assay for the wild type and *tssM* mutant. The strains were inoculated into nutrient and XVM2 containing 0.2% agar. The wild type displayed higher swimming halo in comparison to the mutant strain. The experiment was performed in triplicate and the representative colony morphology of the wild type and mutant strains were photographed.

Figure 3. The transcript level of type IV pili genes was differentially expressed in the *tssM* mutant. (A)

Heat map showing expression of differentially expressed genes related to type IV pili biogenesis. (B) Colony margin characteristics of wild type and mutant strains grown on nutrient and XVM2 containing 1% agar. The wild type exhibited distinct peripheral fringe indicative of type IV pili mediated twitching motility. Magnification scale, 1 mm.

Figure 4. Mutation of *tssM* resulted in dysregulation of genes implicated in various cellular processes.

Transcriptional pattern of differentially expressed genes varying from virulence to metabolism. Heat maps showing the intensity of expression of each gene (rows) expressed as normalized read counts and scaled for each row. Each column represents the samples used in the study and is clustered based on the result of hclust analysis.

Figure 5. Model illustrating the regulatory cascade modulated by *tssM* in AL65 under *in vitro* condition. The proposed model provides an overview into the molecular regulation of different functions in *tssM* mutant ranging from virulence to metabolism controlled by major regulators identified in the present study. One major regulator of virulence factors is the nucleotide secondary messenger c-di-GMP. The intracellular levels of c-di-GMP negatively regulate flagella and T3SS (Hsiao et al. 2011; Su et al. 2016) while have a positive effect on biofilm formation. The higher transcript level of *gum* cluster genes positively regulates the production of exopolysaccharide. In *Xanthomonas* sp. the secretion and delivery of toxic effectors via T3SS is controlled by two major transcriptional regulators HrpG and HrpX (Büttner and Bonas, 2010). Phosphorylated HrpG binds directly to promoter region of *hrpX*. HrpX induce the expression of T3SS and T3Es coding genes by binding to the plant inducible promoter (PIP) box, a DNA motif found in the structural T3SS operon and TSEs gene. The RNA binding protein, CsrA positively regulate T3SS while repress the expression of T4SS, T5SS, T6SS and flagella biogenesis genes. FleQ positively controls flagella biosynthesis and negatively regulates T6SS genes (Wang et al. 2018). The available nitrate from the extracellular space is imported into the cytoplasm by nitrogen-related transporter and reductase coding genes (*narA*, *nirBD*, *nark*, and *amtB*) which are regulated by the master regulator RpoN (σ_{54}). RpoN positively regulate the expression of T3SS and T6SS genes. Under phosphate limited condition the histidine kinase PhoR interact with PstABC-PhoU complex and result in phosphorylation of PhoB which positively regulate the expression of T6SS genes (Chakraborty et al. 2011; Chekabab et al. 2014). At low concentration of zinc, the Zur regulator induces the expression of ZnuABC and activates T6SS genes (Cai et al. 2021). Breakdown of arginine by catabolic enzymes (SpeA and AguA) produce putrescine, exported out of the cell by transporters (PotABC) and helps in ROS scavenging (Solmi et al. 2022).

Fig S1. (A) Biofilm formation of AL65 and *tssM* mutant on abiotic surface quantified by measuring the absorbance (590 nm) of crystal violet. The strains were grown for 24 and 48 hours in XVM2 medium. The biofilm formed on the abiotic surface was stained with 0.1% crystal violet, excess stain was washed with ddH₂O and air dried. The crystal violet stain was solubilized with 40% methanol and the absorbance was measured at 570 nm. The error bar indicates standard deviation. Eight biological replicates of each strain were performed. (B) EPS production by the wild type and mutant strain. The error bar indicates standard

deviation. Three biological replicates of each strain were performed. n.s. denotes not significant.

Reviewer comments:

Reviewer #1 (Comments for the Author):

The work performed by Ramamoorthy and colleagues investigates the global transcriptional changes in *Xanthomonas perforans* driven by deletion of a type-6 secretion system gene, *tssM*. The authors carried out a comparative transcriptomic analysis and used phenotypic experiments (motility, biofilm and EPS levels) to validate the changes in the expression of a few genes. Overall, the manuscript is clearly written, the procedures are properly described, and the results are nicely presented and interpreted. However, despite interesting data showing that the absence of *tssM* globally impacts gene expression, the manuscript fails to provide sufficient novelty. An investigation of the molecular basis for the functional connection between T6SS and any of the molecular processes affected by *tssM* deletion or transcriptomic analyses of cells lacking regulatory genes identified to be upregulated in the *tssM*, organizing the DEGs into groups of genes dependent on each regulatory gene, could add to the manuscript. Below are suggestions for the current version of the manuscript:

We thank this reviewer for the constructive suggestions provided to improve this manuscript. We acknowledge that this study involved only *in vitro* investigation limiting our abilities to derive the inferences related to connecting T6SS and phenotypic traits associated with pathogenesis. However, we believe that this study provides basal level understanding of regulatory mechanisms modulated by *tssM* to further test hypotheses on the role of regulators. The interesting observations made in this study like downregulation of type 3 secretion system gene in *hrp* inducing medium and a possible cross talk between different secretion systems has opened an area for further investigation. These further studies are natural extensions given these derived hypotheses. However, they are beyond the scope of this current manuscript.

Major points:

1. The Results and Discussion sections are very detailed and redundant descriptions of the transcriptional changes observed upon comparing the strains. By adding more data, the description could be shortened to make the text more friendly.

We think that having detailed interpretation for different functional categories as indicated in the manuscript is necessary to provide sufficient context to the readers as to how we proposed the model to explain regulation of T6SS, especially given the intricate nature of the interactions across different secretion systems or pathways.

2. The motility experiment was performed in triplicate, so a plot with mean, error and statistical analysis may be included in addition to the images.

According to the reviewer's suggestion we have performed the statistical analysis for the motility experiment and the result is included in the revised manuscript.

3. RT-qPCR could be used to validate more genes.

The RNA-Seq experiment was performed with four replicates for each time points which provides sufficient dataset to account for any confounding variables. We do not think that RT-PCR is needed in this case to validate the results.

Minor points:

1. Indicate in Fig. 1A the time point (8 and 16h) at the top of the corresponding plot.

The time points have been indicated at the top of the plot.

2. Fig. 2A, 3A and 4. Replace gene number with gene name if possible. This would help the reader interpreting the data, as the text cites gene name, but the figures show gene number in the current version of the manuscript.

The locus tags have been replaced with gene names in Fig. 2A, 3A and 4 in the revised version of the manuscript.

Reviewer #2 (Public repository details (Required)):

They have RNAseq results that should be deposited in a public repository.

The scripts used for analysis, input files, intermediate files and the output files were already publicly available at the time of manuscript submission, along with the Zenodo citation provided for the dataset. I have pasted the data availability section below for your information.

Data Availability

The RNA-Seq reads used in this study are deposited in the National Center for Biotechnology Information sequence read archive database (Bioproject: PRJNA954348). The scripts, input files, intermediate files and output files used for interpretation of the analyses conducted in this work are made available on https://github.com/Potnislabs/T6SS_RNASeq_manuscript as well as [DOI: 10.5281/zenodo.8144145](https://doi.org/10.5281/zenodo.8144145) (117).

Reviewer #2 (Comments for the Author):

The manuscript by Ramamoorthy et al evaluates the transcriptional effects of the mutation in TssM in *Xanthomonas perforans*. They conclude that a crosstalk exists between the regulation of T6SS and that of other secretion systems, especially the T3SS, a major virulence factor in Xanthomonads. The authors perform a thorough analysis of the transcriptional consequences of knocking out this gene, since they report on the differential expression of several functional groups of genes, such as cell wall biosynthesis, stress responses, regulation of gene expression, biofilm formation, phosphorous metabolism. This article provides an opening to the interaction between the T6SS and other systems in the bacterium, something that was suspected before, because of the pleiotropic effects of diverse T6SS mutants, but that was not molecularly tested. They identify some of the candidate genes responsible for this pleiotropic effect and the results that suggest crosstalk between virulence or general signaling systems and T6SS are valuable.

In general, the manuscript is well written and organized in a logical manner, and the conclusions are well supported by the results. However, the major drawback of the manuscript is its extremely descriptive nature. There are a couple of biological supports to the RNAseq analysis,

such as the tests for biofilm formation and motility, but the manuscript does not embark into explaining the discrepancies between phenotypes of T6SS mutants reported in the recent literature. Nor does it attempt to explain why this mutant would be hypervirulent in planta or how the T6SS could contribute to the mentioned microbiome-mediated protection, which are central ideas at the beginning of the document. I believe that the manuscript could use more experiments that connect the expression data with the biological relevance of it in pathogenesis. One major additional concern is experimental: Did the authors measure *in vitro* growth for the mutant and the wild type? If they have different growth rates, is it at all possible that the transcriptional differences observed are due more to the growth stage than to the mutation of the *TssM* gene itself?

We thank this reviewer for the detailed and constructive comments on our manuscript which we believe has improved the scientific merit of our manuscript. The main focus of the present study was to understand the basic regulatory mechanisms modulated by *tssM* using an *in vitro* model system. This study has provided us the preliminary knowledge on the molecular network regulated by *tssM* gene. The goal of this study was not to explain *in planta* phenotype of hypervirulence directly based on gene expression profiles obtained under *in vitro* XVM2 media conditions. Such approach also would not be appropriate given that XVM2 medium only mimics apoplasmic environment and does not take into account the spatial regulation experienced by the pathogen in planta, during epiphytic and endophytic colonization. Thus, while *in vitro* differential expression data provide us with the knowledge of the crosstalk among secretion systems, how they are regulated by the key regulators under *in planta* conditions (for example, quorum sensing c-di-GMP signaling) needs to be understood thoroughly before associating with *in planta* phenotype. With the analysis conducted here, we are now in the position to test some of the key global regulators for their role in mediating crosstalk between T3SS and T6SS under *in planta* conditions during epiphytic and apoplasmic colonization. However, including the results of *in planta* RNA-Seq is beyond the scope of this present study. Instead of additional data, we have mentioned the aspects of the contrasting expression patterns under *in vitro* and *in planta* studies in the discussion section, pasted below.

“In the present study, the expression of genes involved in T3SS and T3E were downregulated in the mutant strain, which was intriguing given the observation of hypervirulent phenotype with mutant in planta (57). We speculate one reason for decreased transcription level of T3SS genes could be the existence of a putative crosstalk between T6SS and T3SS as reported previously. Deletion of single T6SS resulted in decreased expression of T3SS genes in Erwinia amylovora (91), and Pseudomonas syringae pv. actinidiae (92). Recently, Li et al. (93) reported similar results in XVM2 medium where the expression of hrp genes was decreased in Xanthomonas campestris pv. campestris (Xcc) hpaP deletion mutant. HpaP (HR and pathogenicity associated phosphatase), a regulatory protein with both ATPase and phosphatase activity required for complete virulence in Xcc regulating the expression of hrp genes (94). However, a contrasting expression pattern was observed when the pathogen was infected to the plant. Although, the minimal medium XVM2, which is suspected to mimic the plant apoplastic environment, was used as an alternative system for transcriptome expression studies under controlled conditions, it lacks specific plant signals that triggers the hrp regulon in vitro. The induction of T3SS genes is much stronger in the presence of plant-derived signals than in minimal medium (95, 96). In P. syringae DC3000 strain, exposure to tomato exudates activates the expression of hrpK suggesting that the phytopathogen senses the soluble low-molecular weight host signals in the exudate resulting in activation of T3SS genes (97). However, the mechanism by which the pathogen senses the plant signal and the regulatory mechanism that contributes to the induction of host factors and virulence factors in pathogens is unclear. This result and our findings imply that the mechanism of hrp genes activation and regulation under in vitro hrp-inducing conditions is divergent from that in planta. Regardless, the present study provided us with the understanding of the regulatory network of TssM without the complexity of spatial regulation observed in planta. Under such simplified conditions, inactivation of TssM leading to decreased T3SS gene expression suggest intertwined network with direct or indirect interaction of T6SS and T3SS.”

Another major comment raised by the reviewer was regarding the growth defect between the wildtype and the mutant strain. In our previous work, we have performed an in vitro growth experiment using XVM2 medium to measure if there is an inherent growth defect between wild type and the mutant strain. We did not find any difference in the growth rate between wild type and the mutant strain (Liyanapathirana et al. 2022). Hence, we are affirmative that the

difference in transcript abundance was as a result of *tssM* mutation and not because of the difference in growth rate.

Minor comments

The authors should explain more deeply what is known about the role of TssM in the bacterium in the introduction.

In the revised manuscript we have explained in detail the role of TssM in the introduction section.

Please make sure that what the authors describe in the text in the results section corresponds 100% with the results shown in the table, because there are several instances where this was not true. One example is *pilT*, where the authors report that at 16h has an increase in expression, Table S4 reports increase but not significant difference at 8h.

We have checked the manuscript and confirmed that the text corresponds to the table in the revised version.

Some genes do not appear in the supplementary table.

We have thoroughly checked the gene list in the supplementary tables and made sure all the genes are mentioned in the revised version.

There is a lack of statistical support in the biological tests. Could the authors please include these analyses?

We have done the statistics for the phenotypic assays and changed the figures in the revised version of the manuscript.

Please make sure that all references that are needed are cited. Sometimes there is no citation of a reference in important parts of the manuscript.

We have thoroughly checked the manuscript and made sure all the references are cited and also included citation at appropriate places in the revised manuscript.

In addition, we have made edits in the manuscript based on suggestions made by the reviewer in the pdf file.

Re: Spectrum02852-23R1 (Transcriptome profiling of type VI secretion system core gene *tssM* mutant of *Xanthomonas perforans* highlights regulators controlling diverse functions ranging from virulence to metabolism)

Dear Dr. Neha Potnis:

Your manuscript has been accepted, and I am forwarding it to the ASM production staff for publication. Your paper will first be checked to make sure all elements meet the technical requirements. ASM staff will contact you if anything needs to be revised before copyediting and production can begin. Otherwise, you will be notified when your proofs are ready to be viewed.

Sincerely,
Lindsey Burbank
Editor
Microbiology Spectrum